**Investigation**

# Natural variation in the zinc-finger-encoding exon of *Prdm9* affects hybrid sterility phenotypes in mice

Khawla F.N. AbuAlia (ID),[1] Elena Damm (ID),[1] Kristian K. Ullrich (ID),[1] Amisa Mukaj,[2] Emil Parvanov,[2,3,4] Jiri Forejt (ID),[2] Linda Odenthal-Hesse (ID) [1,*]

[1]Research Group Meiotic Recombination and Genome Instability, Max Planck Institute for Evolutionary Biology, Plön D-24306, Germany
[2]Laboratory of Mouse Molecular Genetics, Institute of Molecular Genetics, Czech Academy of Sciences, Vestec CZ-25250, Czech Republic
[3]Department of Translational Stem Cell Biology, Research Institute of the Medical University of Varna, 9002 Varna, Bulgaria
[4]Ludwig Boltzmann Institute for Digital Health and Patient Safety, Medical University of Vienna, 1090 Vienna, Austria

*Corresponding author: Research Group Meiotic Recombination and Genome Instability, Max Planck Institute for Evolutionary Biology, Plön D-24306, Germany.
Email: odenthalhesse@evolbio.mpg.de

PRDM9-mediated reproductive isolation was first described in the progeny of *Mus musculus musculus* (MUS) PWD/Ph and *Mus musculus domesticus* (DOM) C57BL/6J inbred strains. These male $F_1$ hybrids fail to complete chromosome synapsis and arrest meiosis at prophase I, due to incompatibilities between the *Prdm9* gene and hybrid sterility locus *Hstx2*. We identified 14 alleles of *Prdm9* in exon 12, encoding the DNA-binding domain of the PRDM9 protein in outcrossed wild mouse populations from Europe, Asia, and the Middle East, 8 of which are novel. The same allele was found in all mice bearing introgressed *t*-haplotypes encompassing *Prdm9*. We asked whether 7 novel *Prdm9* alleles in MUS populations and the *t*-haplotype allele in 1 MUS and 3 DOM populations induce *Prdm9*-mediated reproductive isolation. The results show that only combinations of the *dom2* allele of DOM origin and the MUS *msc1* allele ensure complete infertility of intersubspecific hybrids in outcrossed wild populations and inbred mouse strains examined so far. The results further indicate that MUS mice may share the erasure of PRDM9$^{msc1}$ binding motifs in populations with different *Prdm9* alleles, which implies that erased PRDM9 binding motifs may be uncoupled from their corresponding *Prdm9* alleles at the population level. Our data corroborate the model of *Prdm9-mediated* hybrid sterility beyond inbred strains of mice and suggest that sterility alleles of *Prdm9* may be rare.

Keywords: reproductive isolation; *Mus musculus*; fertility; asynapsis; *Prdm9*; *Hstx2*; *t*-haplotype

## Introduction

Hybrid sterility is an evolutionary concept of reproductive isolation in which hybrid zygotes develop into healthy adults that fail to produce functional gametes and are thus sterile. In the "Bateson–Dobzhansky–Muller model of incompatibilities", hybrid sterility occurs when 2 or more independently evolved genes are incompatible when interacting within an individual (Bateson 1909; Dobzhansky 1936; Muller 1942). The first hybrid sterility locus identified in mammals was Hybrid sterility 1 (*Hst1*) on chromosome 17 (Forejt and Ivanyi 1974). At the *Hst1* locus, the *Prdm9* gene is responsible for the observed hybrid sterility and encodes PR domain-containing protein 9 (PRDM9) (Mihola *et al.* 2009). The PRDM9 protein (PRDM9) is expressed in testicular tissue and fetal mouse ovaries during the early phases of meiotic prophase I when recombination is initiated (Hayashi *et al.* 2005; Lawson *et al.* 2011). PRDM9 has 3 conserved domains, an N-terminal KRAB domain that promotes protein–protein binding (Imai *et al.* 2017; Parvanov *et al.* 2017; Wang *et al.* 2021), an NLS/SSXRD repression domain with nuclear localization signal, and a central PR/SET domain that confers methyltransferase activity (Powers *et al.* 2016). The C-terminal domain is highly polymorphic and comprises an array of $C_2H_2$-type zinc fingers (ZNFs), which differ among PRDM9 variants in both type and number (Oliver *et al.*

2009; Baudat *et al.* 2010, 2013; Berg *et al.* 2010, 2011; Parvanov *et al.* 2010; Buard *et al.* 2014; Kono *et al.* 2014). Variation among ZNFs is most pronounced in the amino acids at positions −1, 3, and 6 of the ZNF α-helix, which are responsible for recognizing specific DNA target motifs (Billings *et al.* 2013; Baker *et al.* 2014; Walker *et al.* 2015; Patel *et al.* 2017; Altemose *et al.* 2017a, 2017b). Additional amino-acid substitutions in positions −5, −2, and 1 of the α-helix are rarely seen (Parvanov *et al.* 2010; Kono *et al.* 2014). Since only amino acids at positions −1, 3, and 6 are involved in protein–DNA interactions, all other positions are not predicted to affect DNA-binding affinity (Persikov and Singh 2014). Upon interaction with its specific DNA motif, the PR/SET domain of PRDM9 tri-methylates the adjacent nucleosomes on histone-3 by lysine-4 (H3K4) and lysine-36 (H3K36) (Hayashi *et al.* 2005; Wu *et al.* 2013; Eram *et al.* 2014), thereby triggering a cascade of events that initiate recombination, as reviewed in Damm and Odenthal-Hesse (2022).

*Prdm9*-mediated reproductive isolation was discovered in male $F_1$-hybrid progeny of *Mus musculus musculus* (MUS) and *Mus musculus domesticus* (DOM) strains that differ in *Prdm9* alleles (Forejt and Ivanyi 1974; Mihola *et al.* 2009). The PWD/Ph (hereafter PWD) MUS strain possesses the *msc1* allele, and the C57BL/6J (hereafter B6) DOM strain possesses the *dom2* allele. $F_1$-hybrid males from the cross between the PWD female and the B6 male do not complete

chromosome synapsis and spermatogenesis arrests at meiotic prophase I, which prevents them from forming gametes, resulting in postzygotic isolation (Mihola *et al.* 2009; Dzur-Gejdosova *et al.* 2012; Flachs *et al.* 2012). An X-linked locus *Hstx2*, located in a 2.7-Mb region on the proximal part of the X chromosome, modifies the effect of *Prdm9* on the fertility of intersubspecific hybrids. *Hstx2* is structurally distinct between PWD and B6 mice and causes complete hybrid sterility only when the maternal $HstX2^{PWD}$ is active (Bhattacharyya *et al.* 2014; Balcova *et al.* 2016). The interaction between *Prdm9* and the MUS $Hstx2^{PWD}$ allele promotes asynapsis of homologous chromosomes, ultimately leading to meiotic arrest (Bhattacharyya *et al.* 2013, 2014; Balcova *et al.* 2016). In contrast, male $F_1$ hybrids of the reciprocal cross (B6 × PWD) carrying the $Hstx2^{B6}$ allele retain a low level of fertility. The *Hstx2* locus behaves as a recombination cold spot in crosses (Balcova *et al.* 2016), has reduced *Prdm9*-mediated H3K4me3 recombination initiation sites, and lacks DMC1-decorated DNA DSB hotspots (Lustyk *et al.* 2019).

Defective pairing and high levels of chromosomal asynapsis are observed in hybrids with ineffective double-stranded break (DSB) repair (Mihola *et al.* 2009; Davies *et al.* 2016). It has been hypothesized that the molecular mechanism of PRDM9 action is related to the evolutionary divergence of homologous genomic sequences in DOM and MUS subspecies (Davies *et al.* 2016) and, more specifically, to the phenomenon of historical erosion of genomic binding sites of PRDM9 ZNF domains (Davies *et al.* 2016; Forejt 2016; Zelazowski and Cole 2016; Forejt *et al.* 2021). Nucleotide polymorphisms within genomic target motifs may affect the binding affinity of PRDM9 in heterozygous individuals. The preferential formation of DSBs occurs on the haplotype with the motif to which PRDM9 has a stronger binding affinity (Baker *et al.* 2015). However, since the uncut strand provides the template for repair, the less efficient motif is preferentially transmitted to the next generation, which can lead to the erosion of PRDM9 binding sites over time (Jeffreys and Neumann 2002). Therefore, polymorphisms that reduce the binding affinity of a given PRDM9 variant are predicted to become enriched within populations over time, resulting in the attenuation of the hotspots (Boulton *et al.* 1997). Direct evidence for overtransmission has been observed in human and mouse hotspots (Jeffreys and Neumann 2005; Berg *et al.* 2011; Cole *et al.* 2014; Odenthal-Hesse *et al.* 2014). In mouse hybrids, higher-affinity binding sites for a given PRDM9 variant are 4 times more likely to be found on the chromosome of the other species, with which *Prdm9* did not coevolve, suggesting that binding site erosion is a predominant factor driving hotspot loss in several mouse lineages (Smagulova *et al.* 2016). In mice, about 17.5% of hotspots have been eroded in the time it took for the PWD and B6 strains to diverge—averaging to roughly 1 PRDM9 binding site lost every 700 to 1,500 generations (Smagulova *et al.* 2016). Indeed, in sterile hybrids of inbred strains PWD (MUS) × B6 (DOM), heterozygosity for functional PRDM9 binding sites is mainly driven by functional *dom2* binding sites found on the PWD genome that are eroded on the B6 genome and vice versa for *msc1* sites (Davies *et al.* 2016), with recombination being initiated at a large number of asymmetric sets of breaks (Mihola *et al.* 2009; Davies *et al.* 2016). As a result, DNA DSBs initiated at asymmetric sites are difficult or impossible to repair, activating the DNA repair checkpoint or disrupting homolog pairing and synapsis, both of which lead to meiotic arrest. However, fertility was restored when the B6 PRDM9 zinc-finger array was replaced with that of the human PRDM9 variant B, making symmetric recombination hotspots predominant (Davies *et al.* 2016), further supporting the hypothesis

that hybrid sterility is under an oligogenic control, with PRDM9 as the main factor.

Hybrid sterility also occurs outside of laboratory models as *Prdm9* alleles found among MUS and DOM wild-derived inbred strains (Pialek *et al.* 2008) showed fertility disruption in about 1/3 of the intersubspecific male hybrids (Mukaj *et al.* 2020). Mice with *Prdm9* alleles that were closely related to previously identified hybrid sterility alleles showed reduced sperm counts (SC) and low paired testes weights, normalized to body weight (TW/BW) that were associated with high asynapsis rates of homologous chromosomes in meiosis I and early meiotic arrest (Mukaj *et al.* 2020). Replacing $Prdm9^{dom2}$ with the "humanized" targeted $Prdm9^{tm1(PRDM9)wthg}$ (Davies *et al.* 2016) restored fertility in these mouse hybrids, supporting the role of *Prdm9* as the leading player in wild-derived inbred strains (Mukaj *et al.* 2020). Furthermore, although the exon 12 sequence of $Prdm9^{msc5}$ had identical nucleotides in several strains, the degree of fertility reduction observed differed between strains, and the effect of the heterozygosity between genomic backgrounds remained unknown (Mukaj *et al.* 2020).

The relationship between the degree of chromosome asynapsis, meiotic arrest, and the number of expected symmetric DSB hotspots per chromosome was reported in PWD × B6 hybrids (Gregorova *et al.* 2018). Asynapsis was shown to operate in-*cis*, depending on the increased heterozygosity of homologs from evolutionarily divergent subspecies. Introducing at least 27 Mb of sequence homology belonging to the same subspecies (consubspecific homology) rescued the asynapsis of a given autosomal pair (Gregorova *et al.* 2018). $Prdm9^{msc1/dom2}$ also displayed a sterilizing effect in MUS × CAS hybrids, where the rate of synapsis was proportional to the level of non-recombining MUS genetic background (Valiskova *et al.* 2022).

Complete sterility has been observed only in $F_1$ hybrids heterozygous for *Prdm9* alleles *dom2* and *msc1* (Davies *et al.* 2016; Smagulova *et al.* 2016; Mukaj *et al.* 2020). In natural populations, *Prdm9* evolves rapidly, with protein variants behaving like the predator and specific motifs as prey following Red Queen dynamics to avoid negative selection of a complete loss of recombination hotspots over time (Latrille *et al.* 2017; Tiemann-Boege *et al.* 2017). However, while the *Prdm9* gene shows remarkable natural allelic divergence, with more than 150 alleles found in mouse populations to date (Buard *et al.* 2014; Kono *et al.* 2014; Vara *et al.* 2019; Mukaj *et al.* 2020), little is known about how many of these alleles are hybrid sterility-inducing, nor about their DNA-binding motifs.

Furthermore, the fertility of $F_1$ hybrids could be modified by additional hybrid sterility loci. At least 3 autosomal polymorphic hybrid sterility factors exist between PWD and STUS strains (Bhattacharyya *et al.* 2014) and 5 *Prdm9*-dependent quantitative trait loci have been identified in intersubspecific (MUS × CAS) hybrids, segregating on DOM background (Valiskova *et al.* 2022). However, not only do laboratory intercrosses between wild-derived inbred strains differ from the pure form of hybrid sterility observed in (PWD × B6) laboratory crosses, but contrasting patterns are also observed in wild mice. The natural hybrid zone is a relatively recent secondary contact zone across Europe, where only a third of all house mouse males exhibit fertility traits below the range of the pure subspecies. Complex polygenic control of hybrid sterility has been observed, and several interchangeable autosomal loci have been proposed to be sufficient to activate the Dobzhansky–Muller incompatibility in wild mouse hybrids (Dzur-Gejdosova *et al.* 2012; Turner and Harr 2014). A genome-wide association study revealed strong interactions between

Ch17 and Chr X, but most of these loci mapped outside of *Prdm9* and *Hstx2* (Turner and Harr 2014).

Naturally occurring chromosome 17 haplotypes, the *t*-haplotypes (Silver 1985), also strongly influence male fertility in wild mice. They consist of 30 Mb of introgressed sequence transferred from an unidentified *Mus* ancestor in the *Mus musculus* subspecies over 1 million years ago (Hammer and Silver 1993) and encompass the *Prdm9* gene (Trachtulec *et al.* 2008), with the allele most divergent from all other *Prdm9* alleles identified in *M. musculus* to date (Kono *et al.* 2014). Males heterozygous for the *t*-haplotype pass it on to more than half of their offspring, with some variants presenting transmission rates over 90%, while females transmit the *t*-haplotype within the expected Mendelian ratio (Lyon 2003). Despite a strong drive, *t*-haplotypes are only present in 10–40% of all populations of wild house mice, presumably because they also include genes causing male infertility and embryonic lethality (Olds-Clarke 1997; Planchart *et al.* 2000; Schimenti *et al.* 2005; Kelemen and Vicoso 2018). It remains unknown whether *Prdm9* contributes to the observed reduction in fertility associated with *t*-haplotypes.

In summary, the low incidence of sterile wild mouse hybrids in the DOM/MUS natural hybrid zone (Turner *et al.* 2012), together with the reported large number of hybrid sterility loci in intersubspecific backcrosses and intercrosses contrasts with the (PWD × B6) $F_1$-hybrid sterility model based on the *Prdm9* allelic incompatibility, *Hstx2*, and background heterozygosity in PRDM9 binding sites. Further experimental evidence is needed to understand the mechanism of *Prdm9*-driven hybrid sterility and its role in wild mouse populations. Here, we ask whether hybrid sterility is under the *Prdm9* control in wild mice beyond the context of inbred mouse strains. Furthermore, in a simple scenario, one would expect the *Prdm9* sterility-inducing alleles to be ancestral alleles located closest to the common ancestor on the phylogenetic tree. To test this hypothesis, we examined the evolutionary relationship between the known *Prdm9* hybrid sterility alleles and determined the fertility of each newly identified allele in various scenarios and on different hybrid genomes.

## Materials and methods
### Mice
All work involving experimental mice was performed according to approved animal protocols and institutional guidelines of the Max Planck Society and with permits obtained from the local veterinary office "Veterinäramt Kreis Plön" (permit number: 1401-144/PLÖ-004697). Mice, including strains of PWD/Ph strain, C57/Bl6 strain with transgene $Prdm9^{tm1.1}(PRDM9)^{Wthg}$ strain and consomic C57BL/6J-Chr X.1s$^{PWD/Ph}$/ForeJ mice, as well as several wild mice populations were all maintained in the mouse facilities of the Max Planck Institute for Evolutionary Biology in Plön, following FELASA guidelines and German animal welfare law. We analyzed 3 outcrossed populations of DOM mice; first, the French Massif-Central (MCF) population, founded in December 2005 with a starting population size of 16 breeding pairs, with additional wild-caught animals introduced into the breeding population at the beginning of April 2010. These mice were in generation 16 at the start of this experiment, 9 generations since crossing in with the second set of new wild-caught animals. The German Cologne-Bonn (CBG) population was founded in August 2006 with 10 breeding pairs and maintained as an outcross for 14 generations at the start of this experiment. In November 2012, new wild-caught breeding pairs were crossed in. The Iranian population from Ahvaz (AHI) was started in December 2006, with 6

founding breeding pairs. It has been maintained in an outcross for 15 generations at the start of this experiment, after 2 rounds of reduction due to inbreeding depression. Seventeen breeding pairs of mice initially trapped in Almaty, Kazakhstan (AKH) in December 2008, founded the *Mus musculus musculus* population, which had been maintained for 13 generations at the beginning of this experiment. Whole-genome sequencing and transcriptomic data of multiple individuals from each population are publicly available in the European Nucleotide Archive and as custom tracks in the UCSC genome browser (Harr *et al.* 2016).

### Organ withdrawal
Organ withdrawal after euthanasia is not legally considered an animal experiment according to §4 of the German Animal Welfare Act. It, therefore, does not need to be approved by the competent authority (Ministerium für Landwirtschaft, ländliche Räume, Europa und Verbraucherschutz). $F_1$-hybrid males were euthanized after being first rendered unconscious by deliberately introducing a specific $CO_2/O_2$ mixture ratio, then sacrificed using $CO_2$ euthanasia followed by cervical dislocation. To reduce loose hair contaminating the organs during the dissection of the animal, their coat was sprayed with 75% EtOH before organ withdrawal. Spleen, a liver lobe, and both testes were extracted, and epididymides were removed. One epididymis was placed in 500 µl of cold phosphate-buffered saline for sperm counting, and all other organs were immediately snap-frozen in liquid nitrogen and stored at −70°C.

### Fertility phenotyping
We collected body weight (BW) and 2 fertility parameters, paired testes weight (TW) and spermatozoa released from epididymal tissues, counted in million/ml (SC). We normalized testes weight by calculating testes weight to body weight ratios (TW/BW), and sperm cells were counted as follows: One epididymis, including caput, corpus, and cauda, was repeatedly cut in 1 ml of cold phosphate-buffered saline to release spermatozoa. The tube was vigorously shaken for 2 minutes, and spermatozoa in the solution were diluted to 1:40 in PBS. We counted 10 µl of diluted spermatozoa in a Bürker chamber (0.1-mm chamber height), where 2 replicates of 25 squares were counted. In cases when only a few (<10) spermatozoa were found, additional dilutions were prepared and counted. We added the 2 replicated 25 squares counts ($A_{25} + B_{25}$) from spermatozoa released from a single epididymis to approximate spermatozoa released from a pair of epididymides. The epididymal spermatozoa count released in 1-ml PBS was then calculated by taking the paired counts, the volume of 25 squares ($V_{25} = 0.02 * 0.02 * 0.01 * 25 = 0.0001 \text{ cm}^3$), and the dilution factor into account.

### Spreading and immunofluorescence analyses of spermatocytes
Spermatocyte nuclei were spread for immunohistochemistry as described in Anderson *et al.* (1999), with the following modifications. First, a single-cell suspension of spermatogenic cells from the whole testis was prepared in 0.1-M sucrose solution, and then protease inhibitors (Roche 11836153001) were added to the sucrose-cell slurry before dripping it onto paraformaldehyde-treated glass slides. Glass slides were kept in a humidifying chamber for 3 hours at 4°C to allow cells to spread and fix. Slides were briefly washed in distilled water and transferred to pure PBS before blocking in PBS with 5-vol% goat serum. Primary antibodies HORMAD2 (a gift from Attila Toth, rabbit polyclonal antibody 1:700), SYCP3 (mouse monoclonal antibody, Santa Cruz, #74569,

1:50), yH2AX (ab2893. 1:1,000), and CEN (autoimmune serum, AB-Incorporated, 15–235) were used for immunolabeling. Secondary antibodies goat anti-mouse IgG-AlexaFluor568 (MolecularProbes, A-11031), goat anti-rabbit IgG-AlexaFluor647 (MolecularProbes, A-21245), goat anti-human IgG-AlexaFluor647 (MolecularProbes, A-21445), and goat anti-rabbit IgG-AlexaFluor488 (MolecularProbes, A-11034) were used at 1:500 concentration at room temperature for 1 hour. A Nikon Eclipse 400 microscope with a motorized stage control was used for image acquisition with a Plan Fluor objective, 60× (MRH00601). Images were captured with a DS-QiMc monochrome CCD camera and the NIS-Elements program (from Nikon). ImageJ software was used to process the images.

### *Prdm9* genotyping

Ear clips were taken at weaning and used to identify *Prdm9* allelic variation in the wild mouse populations. All $F_1$ and $F_2$ hybrid offspring used in the experiments were instead genotyped from the counted sperm sample taken from one of the epididymides. Furthermore, initial parental *Prdm9* genotyping was confirmed after successful mating (>5 male offspring) by sacrificing all $F_0$ males. All genotyping was done on individual mouse IDs, but in such a way that the experimenter was blind to the matching fertility phenotypes.

### DNA extraction

DNA was extracted from ear clips or whole ears using salt extraction. Briefly, cells were lysed in SSC/0.2% SDS, and proteins were digested using Proteinase K (20 mg/µl), incubating at 55°C overnight. We salted out the DNA using 4.5-M NaCl solution, followed by 2 consecutive rounds of chloroform extraction. The DNA was then ethanol precipitated and washed twice with 70% ethanol, and the pellet was then dried at room temperature and finally dissolved in 30-µl Tris-EDTA pH 8.0. The DNA samples were stored at 4°C for short-term and −70°C long-term storage. The slurry of isolated spermatozoa with epididymal tissues was processed similarly; however, to lyse sperm heads and remove Protamines, we increased the SDS concentration to 1% and added not only Proteinase K (20 mg/µl) but also TCEP (Thermo Scientific 77720, 0.5 M) to a final concentration of 0.01 µM. This extraction method produces a mixture of DNA extracted from somatic and sperm cells.

### Amplification of the minisatellite coding for the zinc-finger array of PRDM9

The ZNF arrays of each mouse were PCR amplified similarly as in Buard *et al.* (2014) on 10–30 ng of genomic DNA in 12-µl reactions of the PCR buffer "AJJ" from Jeffreys *et al.* (1990) using a 2-polymerase system with Thermo Taq-Polymerase (EP0405) and Stratagene Pfu Polymerase (600159) to ensure high-fidelity PCR. When offspring are heterozygous for 2 alleles of different lengths (in most cases), we separated heterozygous bands after gel electrophoresis on Low Melting agarose (Thermo Fischer #R0801) by excising the bands and eluting the DNA using Agarase (Thermo Fischer #EO0461). If 2 heterozygous bands were apparent, excised and eluted product was immediately used in sequencing reactions after estimating the amount of DNA from the gel. If only 1 band was evident, alleles were not separated by size. Therefore, the purified PCR products were cloned using the TOPO TA Cloning Kit for Sequencing (Life Technologies no. 450030), following the manufacturers' specifications before sequencing. We analyzed at least 8 clones per sample.

### Sequencing

Sequencing reactions of either eluted PCR product or picked clones were set up using BigDye 3.0, according to the manufacturer's protocol, then purified using X-terminator, and finally sequenced using 3130x/ Genetic Analyzer. Only PRDM9 variants with less than 12 ZNFs could be sequenced to their ends in both directions. Exon 12 of *Prdm9* was fully sequenced for all alleles; however, forward and reverse sequences overlapped along the entire length of the exon only in alleles smaller than <1,000 bp, such that larger alleles had sequence stretches only covered by either forward or reverse sequencing. Nevertheless, the sequencing products of all alleles still provided sufficient overlap for full-length assembly. We assembled the forward and reverse sequences based on the estimates of fragment sizes from PCR products on gels to accurately assemble the coding minisatellite using Geneious Software (Version 10.2–11). After sequencing and alignment, assembled minisatellites were conceptually translated into the amino-acid sequence of the ZNF domain, and HMMER bit scores were computed using a polynomial SVM (Persikov and Singh 2014).

### Phylogenetic analyses

The phylogeny on all alleles tested for hybrid sterility phenotypes in Mukaj *et al.* (2020), and this publication was computed using the R package "repeatR" from https://mpievolbio-it.pages.gwdg.de/repeatr/ (Damm *et al.* 2022). Briefly, minisatellite-like repeats within the gene are identified, extracted, and filtered for incomplete sequences before matrices based on minimum edit distance (Hamming) were computed using weighting costs $w_{mut} = 1$, $w_{indel} = 3.5$, and $w_{slippage} = 1.75$ as given in Vara *et al.* (2019). These minimum edit distances represent a metric on the set of changes between *Prdm9* minisatellite repeat units of 84 bp in length. As such, it can be used as a measure of genetic distance. We computed 2 distance matrices for each type of repeat, as in Damm *et al.* (2022). The first distance matrix included all nucleotides, while the second matrix excluded nucleotides known to be under positive selection. Two phylogenetic reconstructions of the *Prdm9* hypervariable region were then computed separately from both matrices, using a neighbor-joining approach with the "bionj" function of the R package ape 5.0 (Paradis and Schliep 2019) and rooted on the "humanized" *Prdm9* allele from Davies *et al.* (2016).

### Genotyping of *t*-haplotype Chr17 and X-chromosomal haplotypes near *Hstx2*

The presence of the *t*-haplotype was tested using markers Tcp1 and Hpa-4ps (Planchart *et al.* 2000), and X-chromosomal haplotypes across the refined *Hstx2* interval were tested using primers in Supplementary Table 1 from Lustyk *et al.* (2019). Each forward primer was labeled with either HEX or FAM and amplified using the ABI Multiplex Kit according to the manufacturer's protocol. Fragment lengths were then analyzed by capillary electrophoresis using a 3730 DNA Analyzer. Allele sizes were scored and binned using the Microsatellite plugin in Geneious v.10.2.

### PRDM9 in silico DNA-binding predictions

For in silico DNA motif binding predictions, the nucleotide sequence was first conceptually translated into a protein sequence, and the $C_2H_2$ zinc-finger binding predictions were computed using a polynomial kernel with the method of Persikov and Singh (2014). We converted the matrices from Persikov and Singh (2014) to MEME and JASPAR input files by using the RSAT matrix conversion tool (Santana-Garcia *et al.* 2022) (http://rsat.sb-roscoff.fr/convert-

matrix_form.cgi), choosing the reverse complement option. We used MEME input files for TomTom and JASPAR input files for PWMScan. The thus computed JASPAR files were inputted into PWMScan (Ambrosini *et al.* 2018) to find binding site predictions on the *M. musculus* reference genome mm10 (which most closely resembles the genome of the C57BL/6J strain). The BED files containing the genome-wide putative binding sites of each PRDM9 variant were compared using bedtools intersect (Quinlan and Hall 2010), reporting each incident where bed files overlapped for at least 1 bp.

## Statistical analyses

The majority of graphs, calculations, and statistical analyses were performed using GraphPad Prism software version 9.4.1 for Mac (GraphPad Software, San Diego, CA, USA). Statistical tests are stated in the text and the figure legends. Briefly, pairwise comparisons were performed using unpaired *t*-tests with Welch correction, and as we did not assume equal sample variances a priori, these were compared using *F*-tests. Similarly, multiple comparisons of fertility parameters were first evaluated for differences in sample variances using Brown–Forsythe ANOVA tests. If significant differences between means were observed, we performed Welch's ANOVA with Dunnett's T3 multiple comparisons test. When there was no indication of unequal sample variance, we performed ordinary 1-way ANOVA instead, which we evaluated with Bonferroni multiple comparisons tests. Asynapsis data were compared between genotypes using unpaired *t*-tests with Welch correction, and linear correlation was assessed using the Pearson correlation coefficient (*r*). We tested for transmission ratio distortion using the binomial probability calculator on the VassarStats: Website for Statistical Computation (http://vassarstats.net).

## Results and discussion

Previous analyses revealed 4 alleles of *Prdm9* that induce hybrid sterility in wild-derived inbred mouse strains initially trapped in Europe (Mukaj *et al.* 2020; Forejt *et al.* 2021). We screened additional European wild mice farther away from the hybrid zone and mice from Asia and the Middle East for novel *Prdm9* alleles. Mice were initially caught by Harr *et al.* (2016), founders of the MUS population AKH were initially trapped in Almaty, Kazakhstan (43°16′N, 76°53′E), and founders of DOM populations came from 3 different locations, for the AHI population AHI from the city of Ahvaz, Iran (31°19′ N, 48°42′ E), for population MCF in from the Massif-Central region in France (45°32′N, 2°49′E), and for the CBG population founders were trapped in the Cologne-Bonn region in Germany (50°52′N, 7°8′E). All populations have been housed and maintained as outcrosses for many generations before this study (see *Materials and methods*), and these populations have maintained a much larger genomic diversity than inbred strains despite high degrees of relatedness (Lawal *et al.* 2021). Previous observations of diverse haplotypes in the Iranian basin (Hardouin *et al.* 2015) and demographic analyses of the source populations also confirmed the AHI population as the most ancestral (Fujiwara *et al.* 2022). These outcrossed populations show low levels of introgression between MUS/DOM (Ullrich *et al.* 2017) and moderate levels of bidirectional introgression patterns from *Mus spretus* into all 3 DOM populations (Banker *et al.* 2022). These outcrossed populations of mice with inter-individual genetic diversity have high average SNP densities compared to the C57BL/6 strain, with the number of population-private variants and genomic introgression for each population collected in

Supplementary Table 2. The distribution of original trapping locations of all mice tested for hybrid sterility phenotypes in this study and in Mukaj *et al.* (2020) is shown in Supplementary Fig. 1. Testis mRNA expression levels are available for multiple individuals of each outcrossed population (Harr *et al.* 2016) and have demonstrated robust *Prdm9* expression (Kelemen *et al.* 2022). In summary, these genetically diverse individuals from several outcrossed populations provide a unique resource to evaluate the *Prdm9* allelic incompatibility-mediated hybrid sterility model beyond the context of inbred strains of mice. We screened all populations for individual *Prdm9* alleles by sequencing exon12 of *Prdm9* containing the minisatellite coding for the $C_2H_2$ zinc-finger domain of PRDM9 as described in Buard *et al.* (2014) and Kono *et al.* (2014). The amino-acid variation between individual zinc fingers is shown in Supplementary Figs. 2 and 3. Based on variation in nucleotide repeats and the composition, order, and number of repeats, we have identified 8 full-length *Prdm9* alleles in MUS mice from Kazakhstan, of which *msc7*, *msc8*, *msc10*, and *msc11* were novel, and *msc6*, *msc9*, and *msc12* had been previously observed. The *msc6* allele was previously found in Grozny, Russia, and named Ma8, and the *msc9* allele, previously named Ma12, was found in strains CHD and BLG2 (Kono *et al.* 2014). The *msc12* allele had been found in Illmitz, Austria, and named 7mus1 by Buard *et al.* (2014). All alleles in this study, except the *mmt1* allele, are present only in 1 population, yet some closely resemble alleles of other subspecies. For example, PRDM9[msc11] closely resembled 2 variants from the CAS subspecies, the classical PRDM9[cst1] variant, which possesses serine at position −1 of the alpha-helix of the 8th ZNF (Parvanov *et al.* 2010), instead of the asparagine seen in PRDM9[msc11]. A CAS trapped in Nowshahr, Iran, Ca1 (Kono *et al.* 2014), also differs only by a single amino-acid substitution in position 6 of the alpha-helix of the 6th ZNF, where Ca1 possesses glutamine instead of the lysine seen in PRDM9[msc11]. Alleles *msc7* and *msc10* appear similar and share similarities to MUS 27mus1 trapped in Bulgaria (Buard *et al.* 2014) and CAS Cc4 trapped in Grozny, Russia (Kono *et al.* 2014). The observations of CAS-like alleles in MUS populations are consistent with observations of many "MUS-like CAS" and "CAS-like MUS" samples in genomic datasets that show admixture patterns (Fujiwara *et al.* 2022).

Out of the 6 DOM alleles from France, Germany, and Iran, *dom9*, *dom10*, and *dom11* were novel, and *dom8* from our Ahvaz, Iran population was previously also found in the DOT strain, originally from Tahiti (French Polynesia) and named *16dom1*. We found a single, peculiar *Prdm9* allele in MUS and DOM subspecies in all 4 original trapping locations associated with *t*-haplotypes. We, therefore, tested additional mice with *t*-haplotypes for *Prdm9*, including *Mus musculus castaneous* (CAS) from Taiwan and laboratory mouse strain $T/t^{p4}$ (Forejt *et al.* 1988), confirming that all possessed the same *Prdm9* allele.

Given that a single *Prdm9* allele was found in all mice carrying a *t*-haplotype, regardless of subspecies, we consider it a trans-subspecies *Prdm9* allele *M. musculus t*-haplotype 1 "*Prdm9[mmt1]*". The *mmt1* allele was always heterozygous and occurred in all of our outcrossed populations at high frequencies, together with *t*-haplotypes. We found *t*-haplotypes in 50% of the MUS population from Almaty (Kazakhstan), in 88% of the DOM population from Ahvaz (Iran), in 90% of the mice from Cologne-Bonn (Germany), and 100% of the mice from the Massif-Central (France) population. Initial population frequencies may have gotten heavily distorted due to the *t*-haplotype overtransmission. Thus, the identified alleles may not reflect the initial population frequencies in which they occurred in the wild. However, the

**(a)**

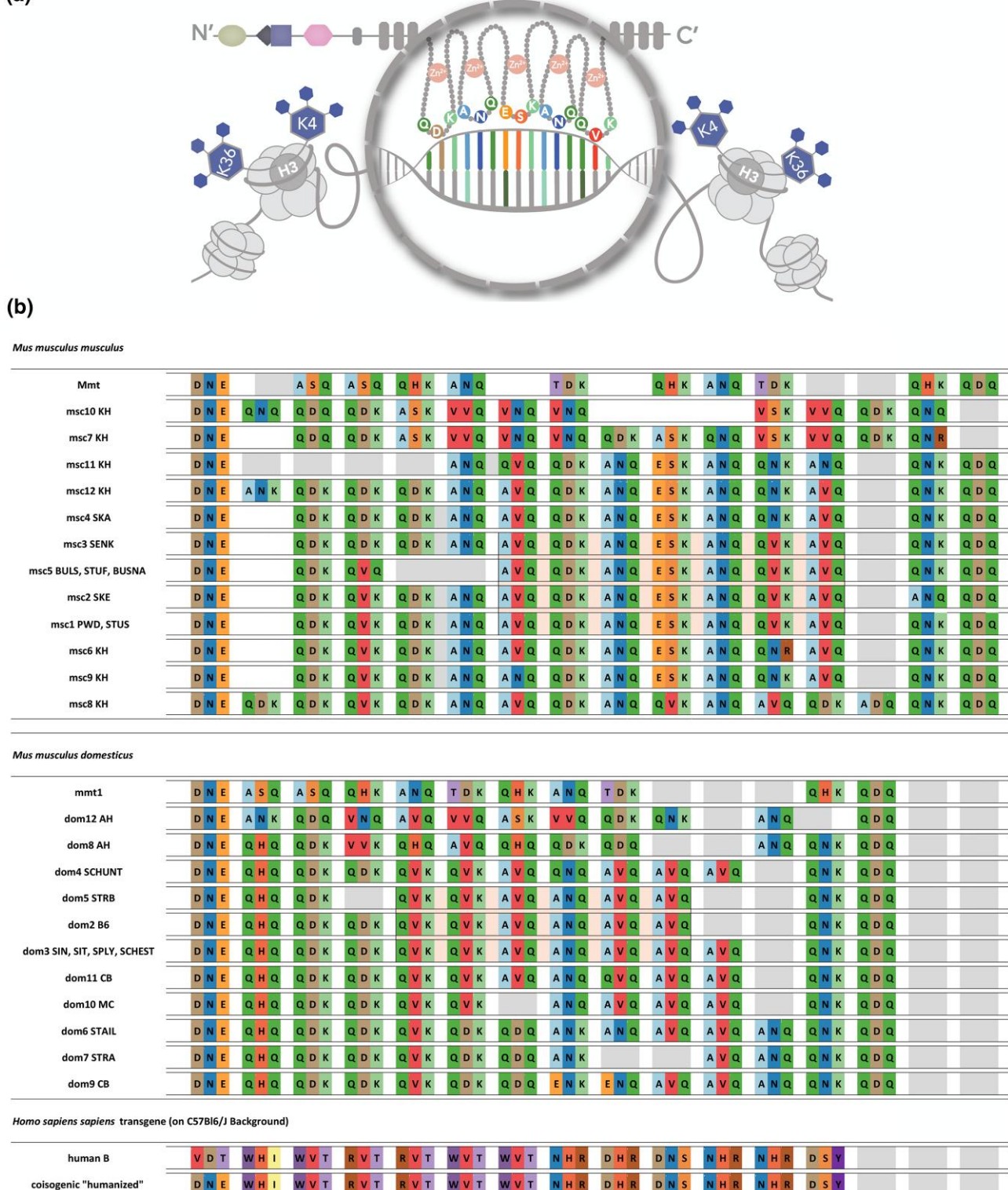

**(b)**

**Fig. 1.** Types of C₂H₂ zinc-finger arrays studied for hybrid sterility phenotypes. a) Cartoon depicting the amino acids in positions −1, 3, and 6 of the alpha-helix of each C₂H₂ ZNFs that are responsible for DNA binding. b) Representation of all C₂H₂ zinc-finger arrays using only acronyms of the amino-acid positions responsible for DNA-binding (as in Supplementary Figs. 2 and 3) zinc-finger arrays of *Prdm9* alleles *msc2*, *msc3*, *msc4*, *msc5*, *dom3*, *dom4*, *dom5*, *dom6*, and *dom7* with alleles from Mukaj et al. (2020).

alleles we identified in the outcrossed population should nevertheless reflect the *Prdm9* alleles in the wild. All alleles identified in this study are shown in Fig. 1, together with previously identified alleles from Mukaj et al. (2020). We named these *Prdm9* alleles

according to the International Committee of Standardized Genetic Nomenclature for mice (MGI), registered them at JAX, and submitted their sequences to GenBank (accession numbers OQ055171–OQ055188).

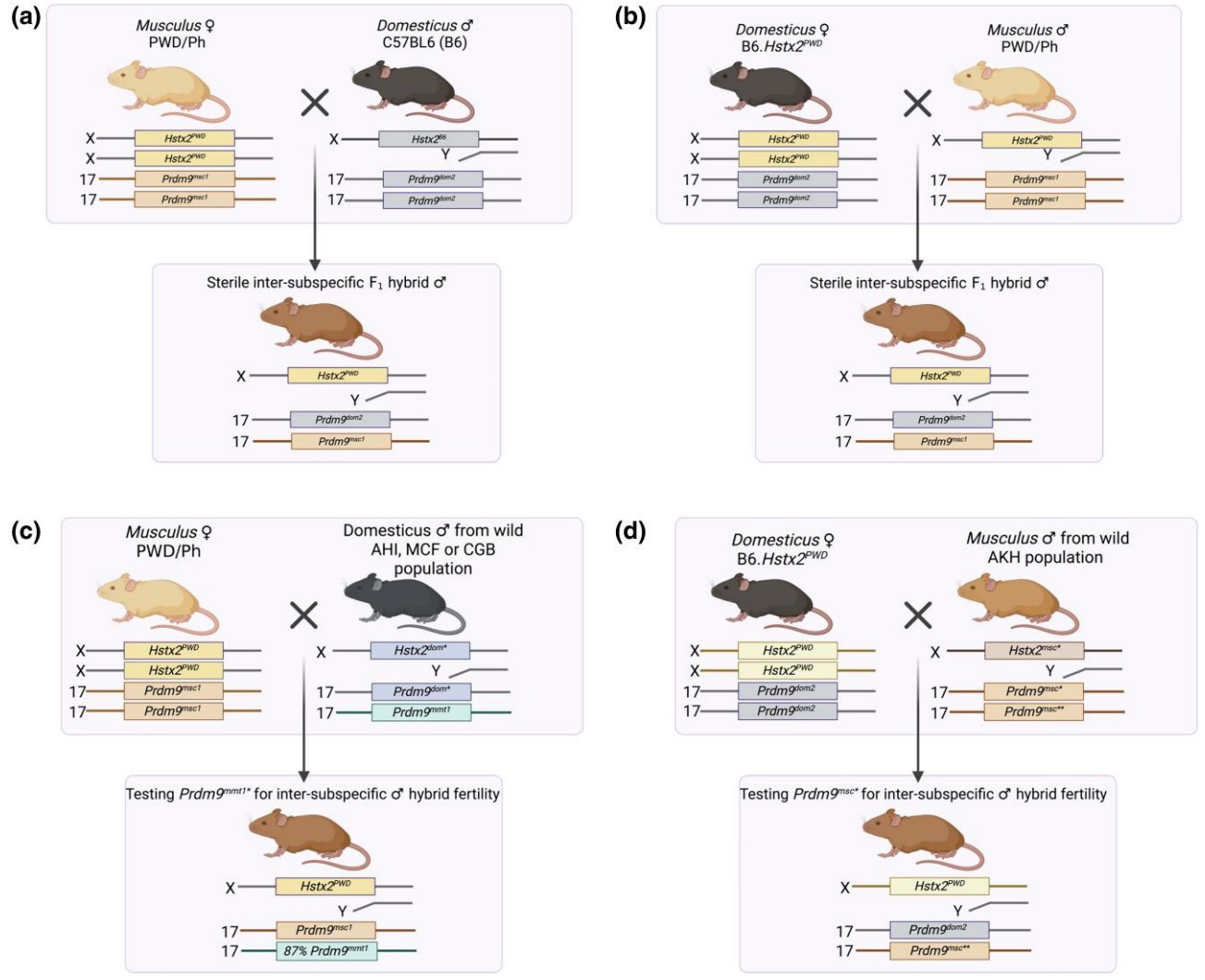

**Fig. 2.** Scheme of intersubspecific crosses and genotypes. a) Classical model for *Prdm9*-driven hybrid male sterility (PWD × B6) and b) reciprocal HS model of B6.*Hstx2*^PWD^ female crossed to B6 male. a, b) Hybrid male sterility depends on 3 main factors: the incompatibility of the PWD and B6 alleles of the *Prdm9* gene (*Prdm9*^msc1^ and *Prdm9*^dom2^), the presence of the PWD (*musculus*) allele of the X-linked *Hstx2* locus, and the MUS/DOM heterozygosity of the F₁ genetic background. c) By substituting the B6 male for the wild DOM male, a change in hybrid sterility can be attributed to the wild DOM *Prdm9* allele. *Prdm9*^mmt1^ was over-transmitted with the *t*-haplotype as all wild DOM were heterozygous for *t*-haplotype. d) By substituting the PWD male for the wild MUS male, a change in hybrid sterility can be attributed to the tester wild MUS *Prdm9* alleles with an unknown effect on hybrid male fertility, *Prdm9*^msc*^, or *Prdm9*^mmt1^ (not shown again in MUS for simplicity). Image created with BioRender.

## Testing *Prdm9* alleles for sterility phenotypes in intersubspecific hybrids

To investigate whether newly identified wild *Prdm9* alleles induce hybrid sterility in intersubspecific hybrids, we adapted the crosses used in previous laboratory models of hybrid sterility to eliminate variation due to the *Hstx2* modifier (Fig. 2). To test the fertility effect of wild DOM alleles, we emulated the PWD × B6 laboratory model (Fig. 2a) but instead crossed PWD (MUS) females with wild DOM males (Fig. 2c). To test MUS alleles, we emulated the B6.DX1s × PWD laboratory model (Mukaj *et al.* 2020) (Fig. 2b) by crossing wild MUS males to C57BL/6J-Chr X.1s^PWD/Ph^/ForeJ females (abbreviated B6.DX1s) (Fig. 2d). B6.DX1s is a consomic DOM strain of C57BL/6J background that carries the *Hstx2* locus within a 69.6-Mb PWD sequence of the proximal end of the X chromosome, essential for F₁-hybrid sterility (Bhattacharyya *et al.* 2014; Balcova *et al.* 2016; Lustyk *et al.* 2019; Forejt *et al.* 2021).

All 9 DOM sires and 5 of the 16 MUS sires possessed *t*-haplotypes. Since the *t*-haplotype is a known meiotic driver

(Lyon 2003; Arora and Dumont 2022; Kelemen *et al.* 2022), we experienced a severe reduction of testable *Prdm9* alleles on wildtype Chr17 in both MUS and DOM mice. Indeed, more than 87% of offspring from fathers with *t*-haplotypes inherited the *mmt1* allele, a significant deviation from the Mendelian 50:50 transmission ratio (2-tailed binomial probability $P \leq 0.000001$, approximated via normal). In contrast, all other *Prdm9* alleles were transmitted at 50:50 in mice without *t*-haplotypes. Since fertility parameters, paired testes weight normalized to body weight, as well as the number of epididymal spermatozoa can differ with the genetic background (Widmayer *et al.* 2020), we first compared physiological variation in the outcrossed source populations of wild and inbred mice (Fig. 3). Wild mice from all 3 DOM populations had comparable sperm counts (Fig. 3a), while testes weights normalized to body weight (TW/BW) were significantly higher in males from CBG and MCF populations than in B6 mice. Similarly, wild MUS had elevated normalized testes weight to body weight ratio compared to PWD (Fig. 3b).

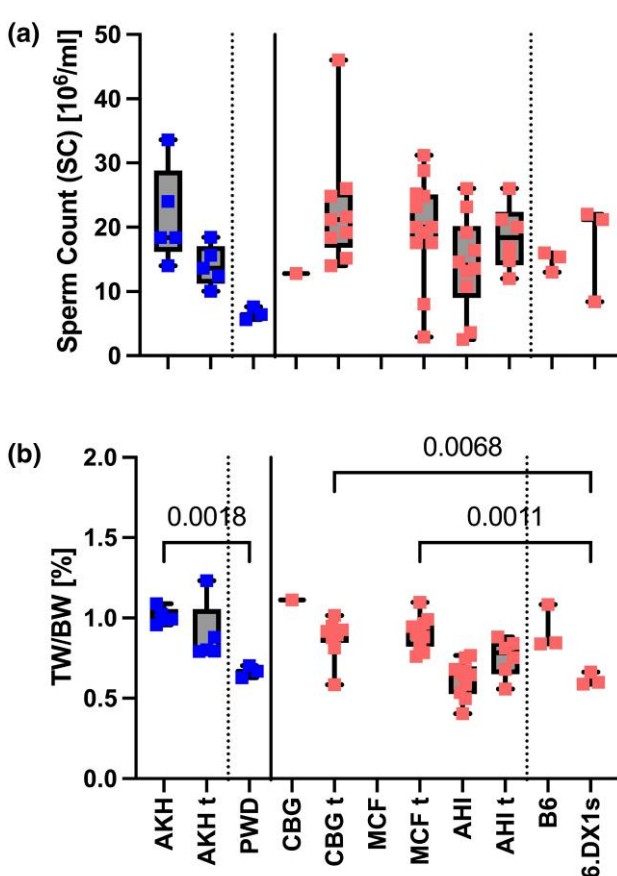

**Fig. 3.** Fertility parameters of wild mouse populations in Ploen, Germany. MUS: AKH, Almaty, Kazakhstan; DOM: AHI, Ahvaz, Iran; CBG, Cologne-Bonn Germany; MCF, Massif-Central France; t, $t^{t/wt}$-haplotype genotype. a) Represents the sperm count for each population. b) Represents the values of the normalized testes weight (TW/BW) ratio for the same populations—we compared fertility parameters between populations using pairwise ANOVA with Bonferroni correction.

The wild mouse males from the same population were almost all heterozygous for different *Prdm9* alleles, and some shared the same *Prdm9* allele. To determine whether the differences in paternal genetic background affected fertility parameters in intersubspecific $F_1$-hybrid males, we compared offspring from different crosses that had acquired the same *Prdm9* allelic combination from different fathers. We observed no effect on sperm counts between offspring from different fathers, except between offspring of 2 fathers from the AHI population, which transmitted t-haplotypes (Fig. 4). However, the TW/BW ratio did differ between hybrids with the *msc8* allele from 2 different fathers as well as between offspring of a t-haplotype carrier from the AKH population (Fig. 4). Because hybrid fertility parameters can also vary with age (Widmayer *et al.* 2020), we restricted our analyses to mice aged 60–120 days and performed regression analyses to exclude age effects on fertility parameters. We saw no effect of age on sperm counts in the tested age range but detected a positive correlation of age with the TW/BW ratio in offspring inheriting the $mmt1^{KH}$ allele and t-haplotypes ($P = 0.0001$) (Fig. 5).

When we pooled fertility parameters of male $F_1$ hybrids by *Prdm9* genotype, the majority had significantly higher TW/BW ratio and SC values than PWD × B6 and B6.DX1s × PWD $F_1$ hybrids (Fig. 6). Exceptions were intersubspecific PWD × DOM hybrids

that inherited the paternal $mmt1^{MC}$ allele and intersubspecific B6.DX1s × MUS hybrids that inherited the paternal *msc11* and *msc12* alleles (Supplementary Tables 3 and 4), whose SC was not significantly elevated compared to *msc1*. The hybrids inheriting paternal *msc12* and *msc11* also showed significantly reduced TW/BW ratio compared to other tested alleles (Fig. 6). Exchanging B6.DX1s females for B6.DX1s .$Prdm9^{hu}$ females, and thus replacing only the PRDM9$^{dom2}$ ZNF array with a human transgenic PRDM9$^{hu}$ ZNF domain (Fig. 6), increased fertility parameters in wild-derived inbred strains that carried sterility alleles (Mukaj *et al.* 2020). We, therefore, performed crosses of B6.DX1s .$Prdm9^{hu}$ females with wild MUS males, but did not observe any significant increase in fertility (Supplementary Fig. 4), presumably because our intersubspecific hybrid males can already be considered (sub)fertile.

Given that all $F_1$-hybrid males were either completely fertile or showed only reduced fertility, we tested whether they displayed chromosomal asynapsis, a hallmark characteristic of *Prdm9*-dependent hybrid sterility (Forejt and Jansa 2023). We immunostained spermatocyte spreads of $F_1$ hybrids that inherited alleles *msc10*, *msc11*, and *msc12*, as well as hybrids that inherited the *mmt1* allele in combination with both DOM and MUS t-haplotypes.

To determine the frequency of pachytene spermatocytes with 1 or more asynapsed bivalents, we immunostained meiotic nuclei with antibodies against phosphorylated histone gammaH2AX and HORMAD2 proteins, which localize to asynapsed autosomes and to the X and Y chromosomes at mid to late pachytene stage (Fernandez-Capetillo *et al.* 2003; Turner *et al.* 2005) (Fig. 6c); we also stained these nuclei with antibodies against the SYCP3 protein, which decorates meiotic chromosome axes. We evaluated 48–113 pachynemas for asynapsis in each individual, scoring each HORMAD2 stained element, except sex chromosomes, as 1 asynapsis event. We determined the percentage of cells with asynapsis and collected all data in the linked Dryad Repository. The $F_1$-hybrid males with wild MUS alleles *msc11* and *msc12* had an elevated proportion of asynaptic pachynemas (*msc11*, 42.4 ± 8.0% and *msc12* 57.2 ± 12.5%), these frequencies are significantly elevated, compared to control mice of pure parental subspecies B6.DX1s and PWD, with asynapsis levels of *msc12* statistically indistinguishable from those of *msc1* (Fig. 6d). In contrast, hybrids with wild mouse allele *msc10* had lower asynapsis, averaging 16.6 ± 8.6%, not significantly different from fertile controls and significantly lower than sterile controls (Fig. 6d). Similarly, mice with t-haplotypes also showed low asynapsis rates (DOM *mmt1* 11.5 ± 12%, MUS *mmt1* 17.4 ± 18.0%). We also tested whether fertility parameters correlated with the meiotic asynapsis rate and observed a weak but significant correlation between high asynapsis rates with both low sperm count (Pearson $R^2 = 0.462$, $P < 0.0001$) (Fig. 6e) and TW/BW ratio (Pearson $R^2 = 0.238$, $P = 0.0046$) (Fig. 6f).

To conclude, the tested *Prdm9* allelic combinations were either completely fertile or showed reduced fertility. Since hybrids with *msc11* and *msc12* alleles had the lowest sperm counts and TW/BW ratio and showed the highest levels of chromosomal asynapsis, we can conclude that *Prdm9* likely drives this effect. The effect is not binomial, either confounded by overall genomic heterogeneity or potential genetic modifiers on the wild genetic backgrounds.

## Role of the wild-derived outbred genetic background in *Prdm9*-driven meiotic arrest

The hypothesis that *Prdm9*-driven hybrid sterility can be explained through the asymmetric erosion of PRDM9 binding motifs was based on the documented erosion of MUS PRDM9$^{msc1}$ binding

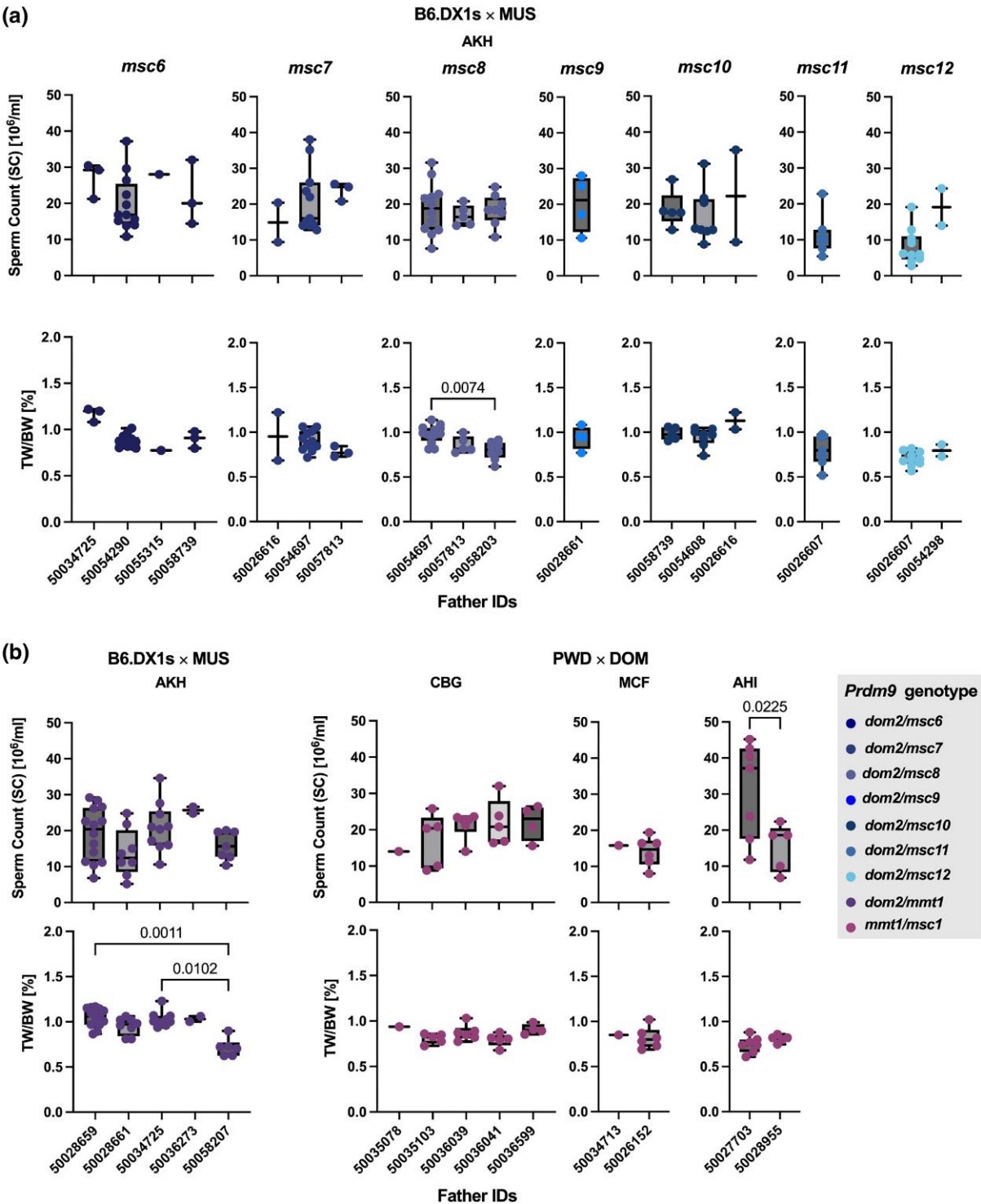

**Fig. 4.** Fertility phenotypes of hybrid offspring grouped by *Prdm9* genotype and sire ID (a) mice with different MUS *Prdm9* from Kazakhstan without *t*-haplotypes. Each graph represents offspring sorted by *Prdm9* genotype, with the father (sire) ID on the *X*-axis. Only the second sire (50054290) was homozygous for *Prdm9*, while all of the others were heterozygous for different *Prdm9* alleles. b) Offspring with *t*-haplotypes is grouped by the source population of the fathers and sire IDs on the *X*-axis. (Left) MUS from Kazakhstan (right), DOM from Cologne-Bonn, Germany (CBG), Massif-Central, France (MCF), and Ahvaz, Iran (AHI). Data pairs were compared using Welch's *t*-test, and multiple comparisons were performed using ANOVA with Kruskal–Wallis' test and corrected for multiple comparisons using Dunn's test. Only significant values are shown on the graph.

sites in the PWD genome and DOM PRDM9$^{dom2}$ sites in the B6 genome (Davies *et al.* 2016). This hypothesis explains the F$_1$-hybrid sterility resulting from heterozygosity for functional PRDM9 binding sites. However, while *Prdm9* can be considered one of the most polymorphic mammalian genes (Oliver *et al.* 2009; Grey *et al.* 2018),

almost nothing is known about the occurrence of PRDM9 binding site erosion in mouse populations or how many alleles have generated significant erosion of their binding sites and whether these eroded sites are stable in wild mouse populations. Following the asymmetry hypothesis, a "fertility" *Prdm9* allele from a male

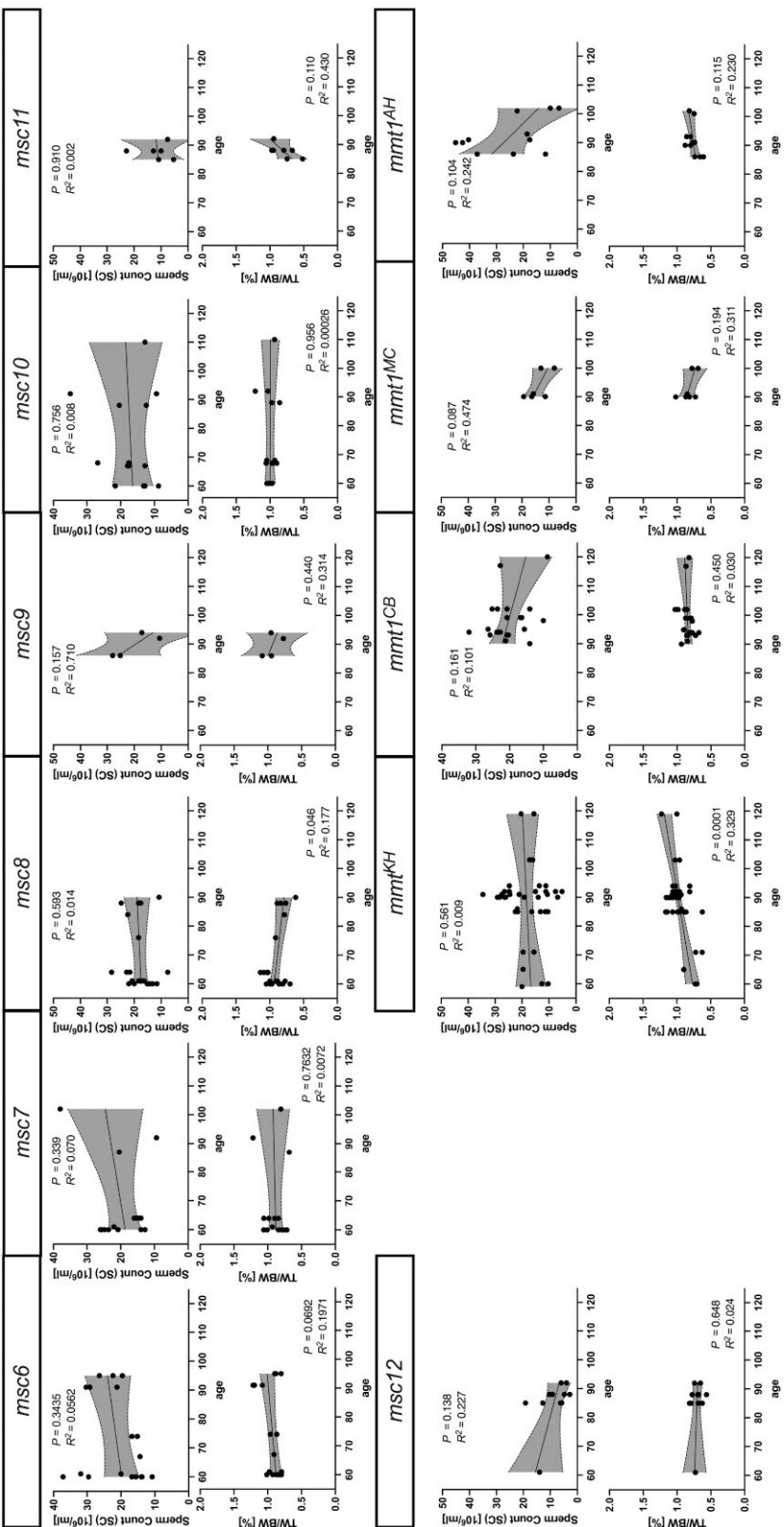

**Fig. 5.** Age correlation with fertility parameters grouped by paternal *Prdm9* allele of F$_1$ hybrids. Each vertical graph represents correlation analyses between age (in days) and fertility parameters for F$_1$-hybrid offspring arranged based on the fathers' allele, sperm count on the top graphs, and normalized testes weight (TW/BW) ratio on the bottom. The same data, as in Fig. 1, were analyzed for correlation with age using simple linear regression with 95% confidence intervals shown.

with no significant erasure of PRDM9 binding sites in the MUS population should produce fertile progeny in (PWD × MUS) × B6 hybrids because erasure of PRDM9$^{msc1}$ sites in the PWD part of

the genome alone is not sufficient to disrupt meiosis in hybrids. However, if the MUS genome with the "fertility" *Prdm9* allele carried eroded *msc1* binding sites, then the fertility of (PWD ×

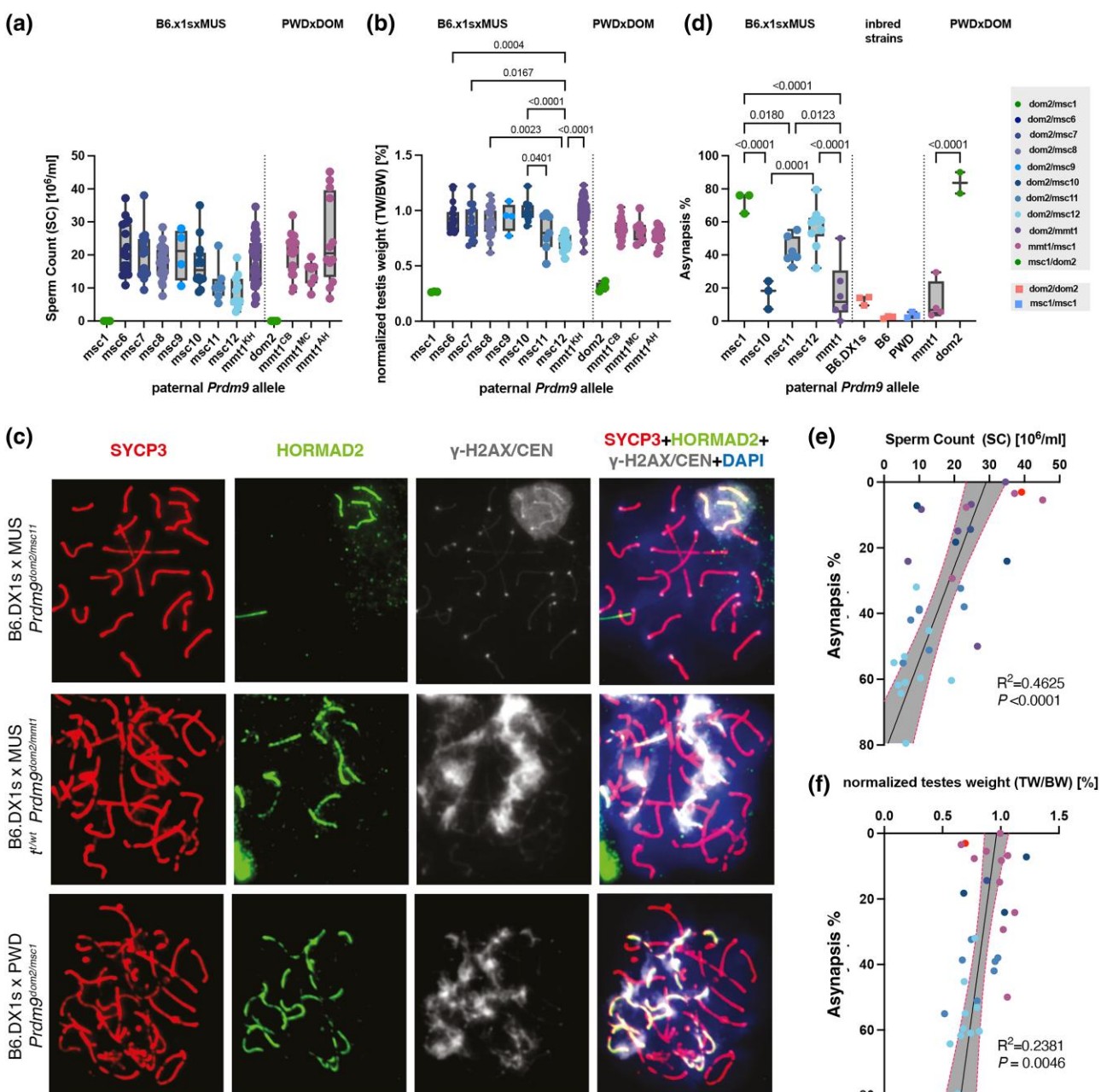

**Fig. 6.** Fertility parameters of intersubspecific hybrids were grouped by *Prdm9* genotype with (a) sperm count and (b) paired testes weights normalized by body weight (TW/BW) tested separately for intersubspecific offspring of (B6.DX1s × wild MUS) or (PWD × wild DOM) intersubspecific crosses (depicted in Fig. 2). We compared their fertility parameters between hybrids grouped by *Prdm9* genotype and also to offspring of (B6.DX1s × PWD) and (PWD × B6), with known hybrid sterility phenotypes using pairwise ANOVA with Bonferroni correction (additional statistics shown in Supplementary Tables 3 and 4). All hybrid males carry the *Hstx2*$^{PWD}$ allele on Chr X. c) The panels show spermatocyte spreads of intersubspecific B6.DX1s × wild MUS hybrids, with differing *Prdm9* genotypes. The defects in chromosome asynapsis were assessed by antibody staining for HORMAD2 protein (green), which marks asynapsed autosomal chromosomes in addition to the nonhomologous parts of XY sex chromosomes that are physiologically observed in normally progressing meiocytes. DNA is counterstained with DAPI (blue). Synaptonemal complex assembly was evaluated by SYCP3 protein immunostaining (red) and the presence of yH2AX (gray). At the zygotene/pachytene transition, clouds of yH2AX mark chromatin associated with asynapsed axes. The localized gray dots represent CEN-labeled centromeres. d) The percentages of asynaptic cells on the Y-axis were grouped by the *Prdm9* genotype on the X-axis. The percentage of asynaptic cells correlated with fertility parameters of intersubspecific F$_1$ hybrids, namely e) sperm count and f) normalized testes weight (TW/BW) ratio, with dotted lines representing 95% confidence intervals, *P*-values, and Pearson $R^2$ values.

MUS) × B6 hybrids would segregate according to *Prdm9*; the *Prdm9*$^{msc1}$ carrying males would be sterile, and their *Prdm9*$^{MUS}$ siblings would be fertile. The same assumption can be made for the outcome of PWD × (B6 × DOM) testcross to test the coupling of eroded PRDM9 binding motif with corresponding PRDM9 allelic zinc-finger domains in DOM populations. To test these alternatives, we generated intrasubspecific (DOM × B6) hybrid males and

crossed them to PWD females to test the DOM-derived *mmt1* and *dom2* alleles against the mixed DOM background (Supplementary Fig. 5). To test MUS alleles and their genetic background, the B6.DX1 females were crossed with intrasubspecific (PWD × MUS) F$_1$-hybrid males. We also performed an analogous cross in reciprocal orientation for MUS alleles, where (PWD × wild MUS) F$_1$-hybrid females were crossed with B6 males. Since

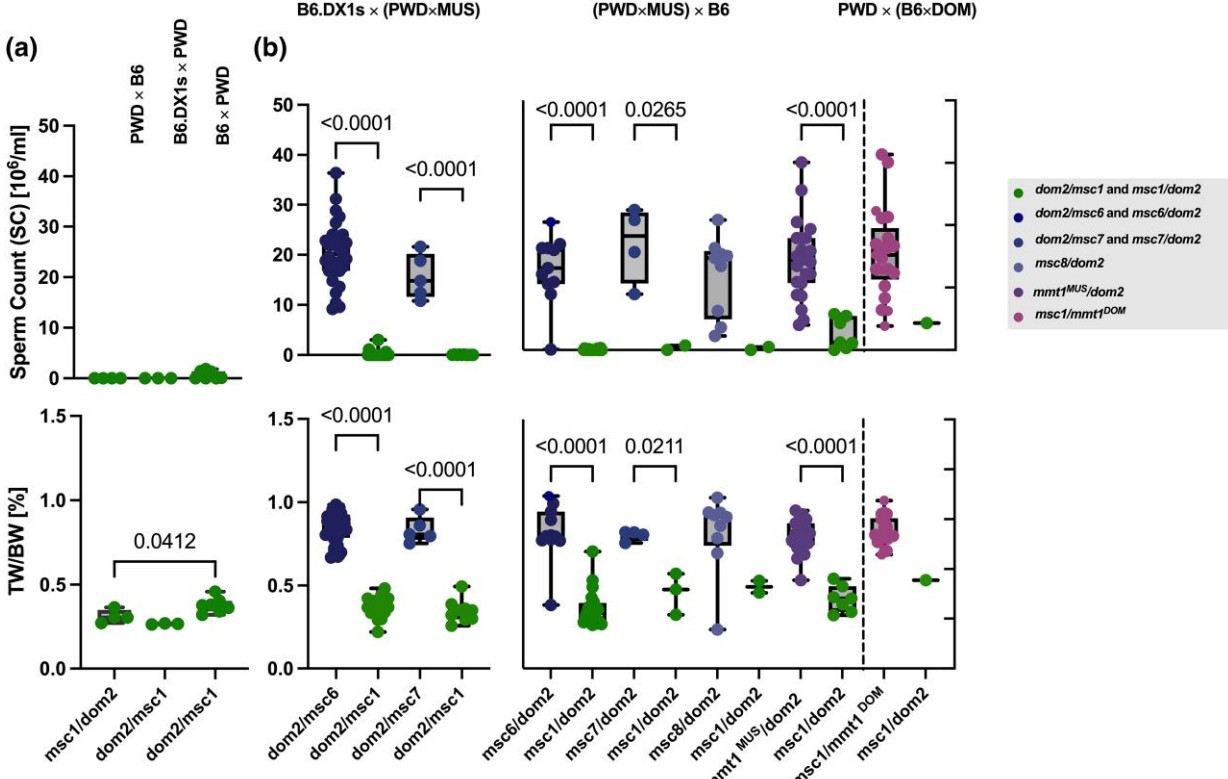

**Fig. 7.** Fertility phenotypes segregate with parental *Prdm9* alleles in reciprocal intersubspecific hybrids. a) Fertility parameters of control cross with the *Prdm9* allelic combination *dom2/msc1* and *Hstx2* allele from PWD or B6. b) Intersubspecific $F_1$ male offspring of (left) B6.DX1s females crossed to intrasubspecific MUS males (middle), intrasubspecific MUS hybrid females crossed to B6 males (right), and PWD females crossed to intrasubspecific DOM males (as shown in Fig. 2). Data were pooled from parents with the same *Prdm9* genotype and compared using pairwise ANOVA with Bonferroni correction, as in Figs. 4 and 5.

not only *Prdm9* but also X-chromosomes segregate in this cross, we only included males with the PWD haplotype containing the refined *Hstx2*$^{PWD}$ locus (Lustyk *et al.* 2019) (see also Supplementary Fig. 6). The testcross results were clear: Siblings who inherited the *Prdm9* alleles *msc6*, *msc7*, *msc8*, or the *mmt1* of MUS or *mmt1* of DOM all showed fertility phenotypes within the physiological range, while their brothers with the allelic combination *msc1/dom2* were sterile (Fig. 7), with TW/BW ratio and sperm count comparable to (PWD × B6) $F_1$ males with the same allelic combination. To conclude, while asymmetric *Prdm9* binding has not been directly demonstrated in our study, due to the infeasibility of performing DMC1 ChIP-seq experiments on wild mouse hybrids, these results nevertheless suggest that mice from MUS and DOM populations with different *Prdm9* alleles may share the same pattern of erased PRDM9$^{msc1}$ (PWD) and PRDM9$^{dom2}$ binding motifs, respectively, suggesting that the erased PRDM9 binding motifs and *Prdm9* alleles are uncoupled at the population level. Furthermore, the partially wild-derived outbred background does not appear to carry additional major genetic modifiers that would prevent sterility per se.

## Variation of X-chromosomal haplotypes in *Prdm9-mediated* sterility

Until now, information on the possible intrasubspecific variation of the *Hstx2* locus has been lacking. Since the interval 65–69 Mb on Chr X containing the *Hstx2* locus appears to be a recombination cold spot (Brick *et al.* 2018; Lustyk *et al.* 2019), we focused on this region in the Kazakh (KH) MUS population and identified 4

different MUS X-chromosomal haplotypes Kha, KHb, KHc, and KHd using only 3 microsatellite markers (see Supplementary Fig. 6) indicating that recombination events within the *Hstx2* locus could have occurred. However, it cannot be excluded that high rates of genome instability at microsatellite markers have led to recurrent mutations masquerading as recombination. We next tested whether these wild X-chromosomal haplotypes differed in the modulation of *Prdm9*-driven hybrid sterility in crosses where *Prdm9* and *Hstx2* segregated. As previously observed with the *Hstx2*$^{PWD}$ sterility allele, fertility co-segregated predominantly with the *Prdm9* genotype, irrespective of which X-chromosomal haplotype the offspring possessed (Supplementary Fig. 7a–c). Regardless of these X-chromosomal haplotypes, all $F_1$ hybrids inheriting any wild MUS *Prdm9* alleles from their mothers were fertile, whereas siblings inheriting the *msc1* alleles were sterile. To conclude, the *Hstx2* did not show functionally defined intrasubspecific polymorphism within the studied MUS population (Supplementary Fig. 7d), and the results did not reveal other genetic modifiers on any individuals' wild genetic background along the maternal germline. An exception to this rule was the *Prdm9*$^{msc1/dom2}$ male offspring, whose mothers were *t*-haplotype carriers, which produced an average of 2.7 M spermatozoa. These males differed significantly in the sperm count from mice with identical genotypes whose mothers did not carry *t*-haplotypes (Mann–Whitney test; SC $P <$ 0.0007) (Supplementary Fig. 7e), suggesting the presence of *Prdm9* fertility modifier(s) in *t*-carrying populations. However, since the *msc1/dom2* allelic combination leads to sterility even on variable genomic backgrounds, we can conclude that hybrid

sterility is indeed under oligogenic control, with *Prdm9* as the leading player.

## Phylogenetic analyses of *Prdm9* alleles in mice

It has been proposed that the role of *Prdm9* in hybrid sterility could be related to the evolutionary divergence of homologous genomic sequences in DOM and MUS subspecies (Davies *et al.* 2016) and, more specifically, to the phenomenon of historical erosion of genomic binding sites of PRDM9 ZNF domains (Baker *et al.* 2015; Smagulova *et al.* 2016) caused by repeated biased gene conversion. Consequently, only the *Prdm9* alleles that have been present for longer evolutionary timescales should generate such partial erosion of their ZNFs binding motifs. To inquire into the evolutionary history of *Prdm9* alleles, we analyzed the phylogenetic relationship of alleles present in our wild mice populations and other alleles for which *Prdm9*-mediated hybrid sterility had been studied (Parvanov *et al.* 2010; Mukaj *et al.* 2020). As an outgroup, the humanized *Prdm9* "B-allele" was added (Davies *et al.* 2016). Since handling sequence repeats is challenging for multiple-sequence alignment algorithms, particularly when the number of repeat units differs, the allelic divergence of minisatellite sequences could not be assessed by standard assembly programs. In addition, genetic distance models (i.e. Tamura and Nei 1993) do not accurately reflect minisatellite evolution driven by de novo recombination between repeats (Jeffreys *et al.* 2013). Therefore, to reflect *Prdm9* evolution more accurately, we applied an algorithm that computes Hamming distances between minisatellite repeats. A Hamming distance is a string metric of the number of substitutions or errors needed to change 1 sequence into another, where all sequences of equal length are vectorized over a finite field. Here we apply it to compare *Prdm9* minisatellite 84-bp repeat units against each other, such that not only point mutations and small indels but also within-repeat-unit processes ($w_{mut} = 1$), as well as repeat-unit insertions and deletions ($w_{indel} = 3.5$) and even repeat-unit duplications and slippage ($w_{slippage} = 1.75$) are taken into account (Vara *et al.* 2019; Damm *et al.* 2022).

We restricted the Hamming metric to high-confidence nucleotide repeats for a more conservative phylogenetic analysis. For each translated amino-acid sequence, we first determined the bit score, a measure of confidence in the homology of a given ZNF to the prediction model, the so-called default Hidden Markov Model (HMMER) gathering threshold. Confidence in a given ZNF is achieved with a bit score above 17.7 for the model used (Persikov and Singh 2014), which was seen for all translated 84-bp repeats, except those coding for the first zinc fingers in the ZNF array, which we found to be conserved in all *M. musculus* *Prdm9* alleles, except *mmt1*.

In the neighbor-joining tree of Hamming distances rooted on the "humanized" *Prdm9* allele, alleles mostly cluster according to mouse subspecies (as shown in Fig. 8). However, not all alleles follow the MUS/DOM subspecies divide. The *mmt1* *Prdm9* allele found in all mice with *t*-haplotypes formed a separate branch irrespective of subspecies and mouse origin, a pattern typical of introgression. The large degree of conservation of *Prdm9* on the *t*-haplotype stands in stark contrast to the rapid evolution of *Prdm9* alleles in mice. Given the remarkable divergence of *Prdm9* alleles in all natural populations studied to date (Buard *et al.* 2014; Kono *et al.* 2014; Vara *et al.* 2019; Mukaj *et al.* 2020), a single ancestral introgression event of *t*-haplotypes into an antecedent of all *M. musculus* subspecies appears more likely than repeated introgression of the same allele at multiple independent events. Furthermore, the frozen pattern of zinc fingers in PRDM9$^{mmt}$ can be explained by a series of naturally occurring inversion blocks, one including the *Prdm9*

locus (Kelemen and Vicoso 2018), likely causing recombination suppression in *t*-haplotypes, therefore constraining *Prdm9* evolution that is mainly recombination-driven (Jeffreys *et al.* 2013).

The *dom12* allele, neighboring a branch of MUS alleles, displays a low divergence to the last common ancestor of MUS and DOM alleles. Except for *mmt1* and *dom12*, all alleles are separated by subspecies origin (as seen in Fig. 8 and Supplementary Fig. 8). The first node separates the *dom8* allele of the Iranian population from all other DOM alleles clustering by subspecies, and a single node leads exclusively to all tested MUS alleles (Fig. 8), which is broadly consistent with the evolutionary history of mice, with the DOM subspecies splitting first with estimated divergence time of 0.130–0.500 million years ago, followed by the CAS and MUS subspecies around 0.110–0.320 million years ago (Phifer-Rixey *et al.* 2020). However, according to our phylogenetic reconstruction, all previously identified hybrid sterility alleles are subspecies-specific alleles of considerable divergence from a common ancestor.

As loci under positive selection can influence divergence times, we calculated a second distance matrix of Hamming distances after removing the hypervariable nucleotides coding for amino acids at −1, +3, and +6 of the alpha-helix, which are responsible for DNA binding (Fig. 1a) before computing a second phylogeny. A few alleles with substitutions outside the hypervariable sites remained separated, pointing to longer divergence times between alleles, the most prominent being mmt1—associated with *t*-haplotypes. In this allele, 9 nucleotides are deleted in the translated amino-acid sequence of the first ZNF of the array, removing 3 amino acids, including 1 of the zinc-binding cysteine ligands (Supplementary Fig. 2). In addition, distinct amino acids are seen in positions −1, +3, and +6, not present in any other PRDM9 variants, such as TDK and ASQ, with additional differences between the 2 types of ASQ ZNFs in positions −8 and +5. Similarly, the ANQ ZNF found in mice with *t*-haplotypes differs from the ANQ found in mice without *t*-haplotypes at position −2, a pattern previously seen in mice with *t*-haplotypes (Kono *et al.* 2014). However, while the *mmt1* allele remained separated, some subspecies-specific nodes disappeared for most alleles where larger divergence times are not driven by positive selection alone. Here, the tree's topology changes dramatically (compare Fig. 8 with Supplementary Fig. 8). Some repeats possess nucleotides coding for a tryptophan (W) residue in position −5. Tryptophan's nonpolar, aromatic, and neutral chemical properties differ from the arginine (R) typically found in this position, which is polar and strongly basic. Secondly, there are 2 types of last repeats; the rarer one codes for arginine (R) in position 13, while the more common type possesses nucleotides coding for aliphatic and nonpolar glycine (G) (Supplementary Fig. 2). The nucleotides coding for glycine in this position appear to be the ancestral alleles, as the same amino acid is also encoded at this position in human PRDM9, included in the genetically engineered "humanized" *Prdm9* allele B in mice, where it can rescue sterility (Davies *et al.* 2016) (Supplementary Fig. 3). The closely related alleles to the common ancestor of *M. musculus* *Prdm9* alleles are *msc11*, *msc5*, and *msc10*, which are now neighboring DOM alleles, possibly placing their origin before a clear separation into MUS and DOM subspecies. Curiously, while the full-length sequence of the *msc7* allele had previously appeared most closely related to the *msc10* allele (Fig. 8), it is now found neighboring the *msc2* allele from SKE/JPia, within a tree of subspecies-specific alleles (Supplementary Fig. 8). In comparison, most alleles differed only at hypervariable sites and are therefore found on the same branch once loci under positive selection are removed. They include, on

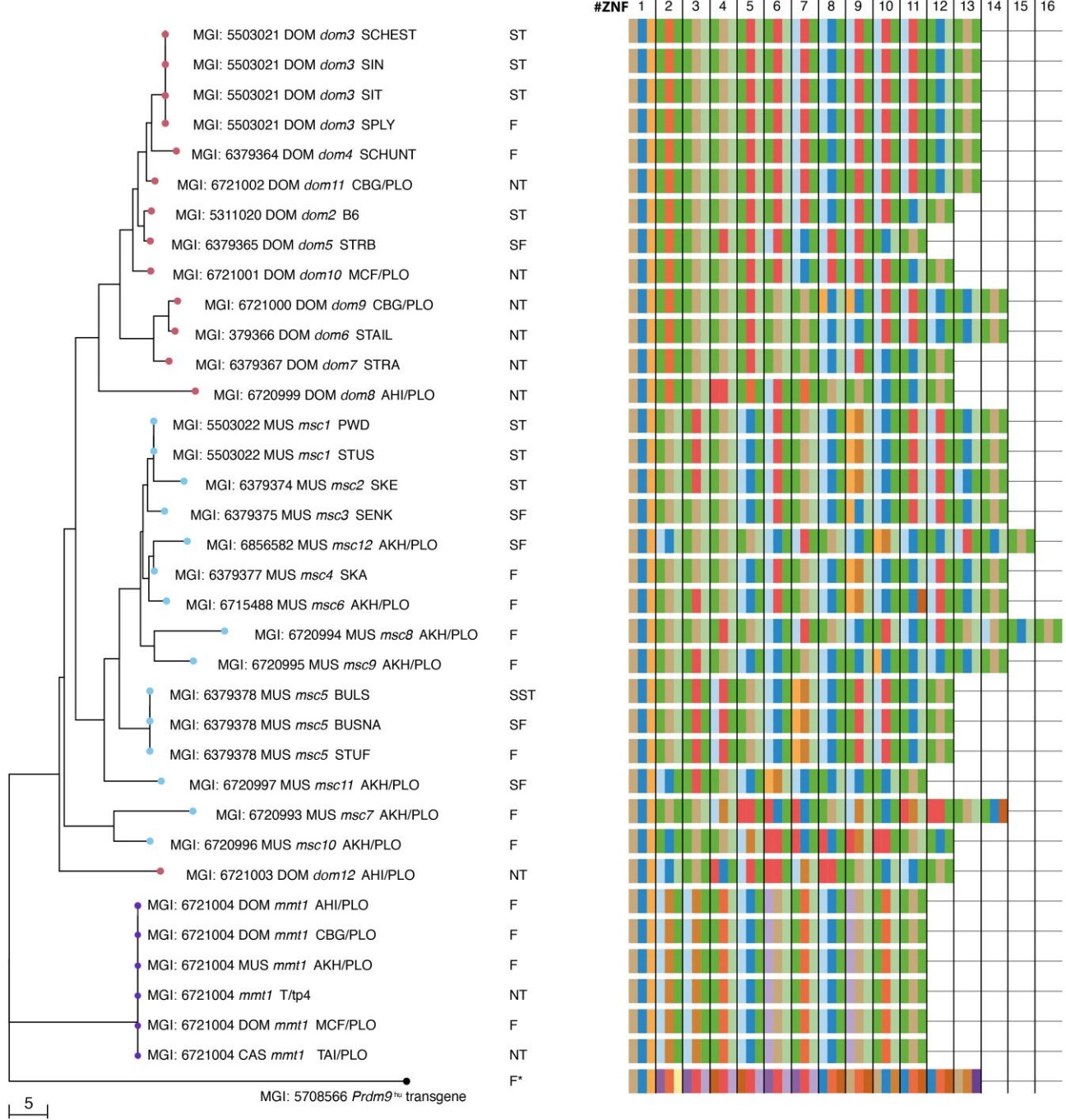

**Fig. 8.** Neighbor-joining tree of the *Prdm9* exon12 minisatellite, which encodes the DNA-binding domain of PRDM9 calculated on the nucleotide sequences of all *Prdm9* alleles in this study and Mukaj *et al.* (2020) that code for the $C_2H_2$ ZNFs array, with red nodes for DOM and blue nodes for MUS alleles, and with purple nodes depicting the *t*-haplotype allele found in MUS and DOM. To the right of the tree, a table depicts the $C_2H_2$ ZNF array encoded by each allele, with boxes (colored as in Fig. 1, Supplementary Figs. 2 and 3) representing only the amino acids responsible for the DNA contacting of each ZNF.

the one hand, *msc1*, *msc4*, and *msc9*; on the other hand, *dom6* and *dom9*. The divergence between *Prdm9* alleles thus appears predominantly driven by positive selection on the hypervariable sites.

In conclusion, the complementary phylogenetic analyses support the accelerated evolution of the hypervariable DNA-binding sites of the PRDM9 zinc-finger array and reiterate an evolutionary history in which *M. musculus* originated in Asia and the Middle East before dispersing across Europe. The phylogenetic analyses

further support a scenario in which MUS and DOM have split recently and are still speciating but do not reveal any apparent clustering of alleles co-inducing hybrid sterility by subspecies. Admittedly, no evidence was found to support the idea that the hybrid sterility susceptible alleles (*msc1*, *msc1*, *msc5*, *dom2*, *dom3*, and *dom5*) belong to the evolutionary oldest ones closest to the common ancestor. On the contrary, the *msc1*, *msc2*, and *dom3* alleles are the most distal, sitting on the most distant branch of the phylogenetic tree (Fig. 8 and Supplementary Fig. 8).

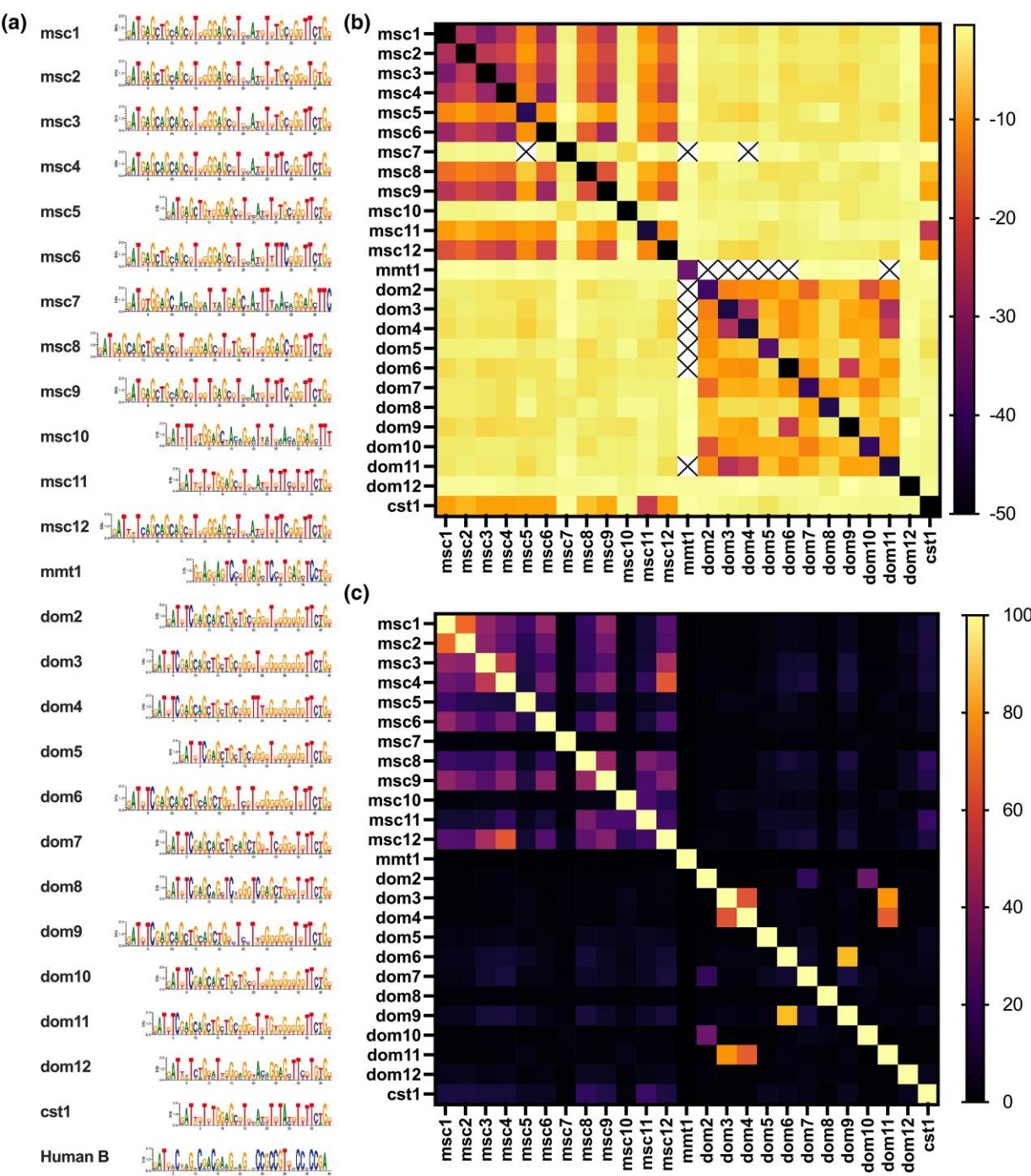

**Fig. 9.** In silico predicted PRDM9 DNA binding. (a) PRDM9 DNA-binding motifs are represented as sequence logos of the underlying positional weight matrices, which were predicted using the polynomial kernel method by Persikov *et al.* (2009) and Persikov and Singh (2014) on translated nucleotide sequences of alleles in this study, and Mukaj *et al.* (2020). (b) Motifs were compared using TomTom, within the MEME suite, which computes the probabilities that a random motif would be better matched than the input motif. TomTom output *P*-values were log-transformed and are shown in a heatmap matrix, such that darker colors represent better matching of sequence motifs, and lighter colors represent weaker similarities of motifs, with crossed-out values representing incidences where no similarity was found. When the motif is compared to itself, the probability of another motif binding better than the motif itself is zero in most cases; however, as values of zero cannot be log-transformed, these incidences are represented by black boxes. c) Overlap of genome-wide binding sites, predicted for each PRDM9 binding motif, with brighter colors showing more extensive genomic binding site overlap between different DNA-binding motifs.

## Comparative analyses of DNA-binding motifs

The phylogenetic approaches are based on coding-nucleotide sequences; however, PRDM9 was identified as a candidate meiotic regulator based on a sequence motif enriched in human recombination hotspots but not in primate recombination hotspots (Myers *et al.* 2010). To inquire into the DNA-binding motif, we predicted DNA-binding motifs in silico (Fig. 9a), using the polynomial SVM prediction method for each conceptually translated PRDM9 ZNF (Persikov and Singh 2014). To inquire whether similar coding sequences of the PRDM9 ZNF array would predict binding to the

same or highly similar motifs, we used TomTom (Gupta *et al.* 2007). Indeed, many DNA-binding motifs are highly similar (Fig. 9b), including predicted DNA-binding motifs of ZNF domains encoded by *msc1*, *msc2*, *msc3*, *msc6*, *msc9*, and *msc12* in MUS, or *dom6* and *dom9*, *dom3*, *dom4*, and *dom11* in DOM. Highly similar DNA-binding motifs of differing ZNF domains may be able to activate the same hotspots. Indeed, cross-activation of the same hotspot by PRDM9 variants encoded by several highly similar alleles has been observed in human hotspots (Berg *et al.* 2010, 2011). Likewise, highly similar predicted DNA-binding motifs were also enriched in contemporary mouse meiotic recombination hotspots, even in other mouse strains (Smagulova *et al.* 2016). To investigate putative genome-wide targets of each predicted DNA-binding motif, we used PWMScan (Ambrosini *et al.* 2018). We then quantified how many genomic targets of predicted binding sites would be shared between PRDM9 ZNF domains of different alleles. Considerable overlap in genome-wide putative binding site distribution is seen particularly across highly similar alleles (Fig. 9c). Exceptionally high overlap of putative genomic binding sites is observed for ZNF domains encoded by MUS HS alleles *msc1* and *msc2*, *msc3*, as well as between *msc4* and *msc12*, and to a lesser extent between *msc6* and *msc1*.

## Conclusion

In summary, none of the 7 novel allelic combinations of wild MUS *Prdm9* alleles produced completely sterile $F_1$-hybrid male offspring in combination with the *dom2* sterility allele, which is consistent with the low incidence of completely sterile hybrids reported in the wild (Turner *et al.* 2012). Instead, we saw either completely fertile intersubspecific hybrids (*dom2* in combination with *msc6*, *msc7*, *msc8*, *msc9*, or *msc10*) or a significant *Prdm9*-dependent reduction of fertility and increased levels of meiotic asynapsis (*dom2* in combination with *msc11*, or *msc12*). Thus, combined with the previous data from wild-derived inbred lines (Mukaj *et al.* 2020), it appears that sterility alleles of *Prdm9* may be rare. While the data on *Prdm9* polymorphisms in wild house mouse populations are accumulating, and indeed, the *Prdm9* genes show remarkable natural allelic divergence, with more than 150 alleles characterized in mouse populations to date (Buard *et al.* 2014; Kono *et al.* 2014; Vara *et al.* 2019; Mukaj *et al.* 2020), little is known about their DNA-binding motifs and their degree of erosion. Although our study did not directly demonstrate asymmetric *Prdm9* binding, because DMC1 ChIP-seq experiments on wild mouse hybrids are not feasible, these results nonetheless provide evidence of eroded *msc1* hotspots in 5 populations with fertile *Prdm9* alleles.

This was surprising, as it suggests a decoupling of the evolutionary dynamics of the PRDM9 zinc-finger domains and their binding sites. In other words, the erosion of the *msc1* hotspots may be much more common in natural populations than the *msc1* allele itself. This would align with the observation that recombination maps based on linkage disequilibrium revealed a significant overlap of historical hotspots of the AKH population with contemporary *msc1-activated* hotspots of the PWD strain (Wooldridge and Dumont 2022). At the same time, there is weak conservation of recombination maps at the broad and the fine scale, and most hotspots are unique to the AHI, MCF, CGB, and AKH populations (Wooldridge and Dumont 2022). The other unanswered question relates to the evolutionary age of the *Prdm9* sterility alleles. In a simple scenario, the sterility-inducing alleles would be expected to be the oldest, situated closest to the common ancestor on the phylogenetic tree. Still, the analysis revealed

the opposite: *msc1*, *msc2*, and *dom3* are the most distal, and therefore most likely the youngest, alleles.

Another question concerns the enigmatic *t*-haplotypes present in all 3 major mouse subspecies and carrying, in all examined cases, the same *Prdm9* allele coding for the same zinc-finger domain. Is it so old because it arose before the ancestral species split into the 3 subspecies? If so, why does it not behave as a sterility allele? Where did it come from if it is a recent introgression due to extremely high transmission distortion? The structure of ZNF and other sequences of *t*-haplotypes shows no similarity to any extant subspecies. Clearly, more experimental evidence is needed to understand the evolutionary dynamics of *Prdm9*-driven hybrid sterility.

## Data availability

Nucleotide sequences of *Prdm9* alleles were deposited to GenBank (accession numbers OQ055171–OQ055188). The fertility datasets generated and analyzed during the current study, DNA-binding predictions of $C_2H_2$ zinc-finger domains encoded by each allele, as well as their genome-wide DNA-binding predictions, are available in a Dryad repository under DOI 10.5061/dryad.bzkh189cm. The R package to calculate the genetic distance between complex repeats is available at https://gitlab.gwdg.de/mpievolbio-it/repeatr.

Supplemental material available at GENETICS online.

## Acknowledgments

We are grateful to Peter Donnely for the transgene $Prdm9^{tm1.1}(PRDM9)^{Wthg}$ ("humanized") strain and Attila Toth for the HORMAD2 antibody. We thank Christine Pfeifle and the entire mouse house team at the MPI in Plön for their help with mouse breeding and maintenance. We are grateful to Heike Harre for help with fertility phenotyping, Vladana Fotopulosova for help with cytology, and Nicole Thomsen for help with DNA extraction and genotyping.

## Funding

First, we would like to thank the Max Planck Society, the German Research Foundation (DFG) (grant no. OD112/1-1 to LO-H), the DAAD (57334341 to JF and LO-H), and the Czech Science Foundation (grant no. 22-299-28S to JF) for funding.

## Conflicts of interest

The author(s) declare no conflicts of interest.

## Author contributions

LO-H, KFNA, ED, KKU, and AM acquired, analyzed, and interpreted data. JF, EP, and LO-H conceived the project, designed the work, supervised data acquisition, and analyzed and interpreted the data. LO-H wrote the manuscript. All authors have read and approved the final manuscript.

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

*Editor: A. MacQueen*