## [Peer Review File · Genetics]

Natural variation in the zinc-finger-encoding exon of *Prdm9* affects hybrid sterility phenotypes in mice

Khawla Abu Alia, Elena Damm, Kristian Ullrich, Amisa Mukaj, Emil Parvanov, Jiří Forejt, and Linda Odenthal-Hesse

NOTE: The reviews and decision letters are unedited and appear as submitted by the reviewers.

In extremely rare instances and as determined by a Senior Editor or the EIC, portions of a review may be redacted. If a review is signed, the reviewer has agreed to no longer remain anonymous.

The review history appears in chronological order.

Review Timeline:

Submission Date:	2023-01-16
Editorial Decision:	2023-02-15
Resubmission Received:	2023-09-06
Editorial Decision:	2023-10-02
Resubmission Received:	2023-11-20
Editorial Decision:	2023-12-08
Revision Received:	2024-01-04
Accepted:	2024-01-05

February 15, 2023

GENETICS-2023-305874

Natural variation in *Prdm9* affecting hybrid sterility phenotypes

Dear Dr. Odenthal-Hesse:

Three experts in the field have reviewed your manuscript, and I have read it as well. All the reviewers appreciate that your study addresses the molecular underpinnings of speciation, which is an important topic. However, some reviewers and I struggled to understand what novel results or significant insight your study provides. Two reviewers furthermore list concerns about conclusions made in the manuscript that are not rigorously supported by the data provided. While your manuscript is not currently acceptable for publication in GENETICS, we would re-review a substantially revised manuscript if the reviewer concerns can be resolved.

All reviewer comments should be addressed in a revised manuscript; you can read their reviews at the end of this email. It is particularly important that your revised manuscript draws rigorous conclusions from the presented data, which may require removing or dialing back certain conclusions in the narrative, or addition of new data to strengthen an interpretation.

It is also important that your revised manuscript clearly communicates what question(s) or hypotheses are addressed by the study, and what new insight the presented findings provide. The reviewers furthermore point out numerous areas of the narrative that require editing or additional explanation for the manuscript to be clearly understood by the reader.

If you choose to resubmit a substantially revised version of your manuscript, please let the editorial office know approximately how long you expect to need for revisions.

Upon resubmission, please include:

1. A clean version of your manuscript.
2. A marked version of your manuscript in which you highlight significant revisions carried out in response to the major points raised by the editor/reviewers (track changes is acceptable if preferred).
3. A detailed response to the editor's/reviewers' feedback and to the concerns listed above. Please reference line numbers in this response to aid the editor and reviewers.

Your paper will be sent back out for evaluation by at least two of the reviewers.

Additionally, please ensure that your resubmission is formatted for GENETICS
<https://academic.oup.com/genetics/pages/general-instructions>

Follow this link to submit the revised manuscript: <https://genetics.msubmit.net/cgi->

bin/main.plex?el=A6NR6FLk5A6NJF7I6A9ftdf8jzsFU3sTtcxeKeNzC1QZ

Sincerely,

Amy MacQueen
Associate Editor
GENETICS

Approved by:
Jeff Sekelsky
Senior Editor
GENETICS

Reviewer #1 (Comments for the Authors (Required)):

The authors look at PrdM9 alleles in different genetic backgrounds to characterize novel prdm9 alleles occurring in natural populations and to assess their effect on fertility.

Overall, this is a nice manuscript. Studies such as these take a long time, and the authors did do experiments carried out in backgrounds with Prdm9 alleles that haven't been looked at before. However, I am struggling to see what this manuscript teaches us, that hasn't been shown before.

I have a number of comments.

Line 202 of methods, a character is missing.
Line 260, "to ensure d high-fidelity", "d" is an error

Line 273: "Only PRDM9 variants with less than 12 ZNFs..." How many alleles are expected to be larger than 12 ZNFs? How many alleles are lost due to this constraint?

Line 345-6: "... human Prdm9 allele B, able to rescue sterility when genetically engineered into mice"

This is meant to say that the Prdm9 allele B rescues sterility, not that the ancestral Glycine alone can rescue?

Line 374: to clarify, since the t-haplotype frequency took over the mouse populations while housed in the lab, the prdm9 alleles identified do not reflect the prdm9 variants in wild

populations?

Line 376-393: Can you speak to how these amino acid changes are predicted to affect DNA binding specificity?

Line 410: "erosion of PRDM9 binding sites results in asymmetric initiation when..." asymmetric initiation of what? Presumably meiotic recombination.

Line 557: how did you determine spermatocytes were in pachytene, and not late zygotene or early diplotene

Line 651-654: since t-haplotype is known to have transmission ratio distortion in males, isn't it impossible to say that your observations are due to prdm9 fertility modifies? Can you say this for Prdm9 alleles that are not associated with t-haplotype? And if so, wouldn't it be better to make that argument with these alleles.

Figure 1: Can you add the number of ZFN repeats?

Explain what the colors of each dot next to the allele means. For example, MGI:5503021 Mmd dom4 SCHEST has a red color. What does read mean?

Somewhere, can you speak to the amount of variation between the mouse strains you are looking at. Like, dom strains contain X number of diverged nucleotides per 1KB, or something, to get an idea of how, beyond phylogeny, diverged the genomes are?

Line 956: some error in this line of text

I've noticed that the testis weights do not always correlate to sperm counts. This either means, testis weight is a bad parameter for fertility, or I suspect more likely, you did not control for body weight. If you do average testis weight normalized to animal body weight, I expect that testis weight will correlate with sperm counts/fertility much better. Otherwise, id reconsider using testis weight and look at sperm counts only

Figure S2/3: what does the yellow shade and pink shade mean?

Line 1010: there is some error in this line of text

Reviewer #2 (Comments for the Authors (Required)):

This paper describes a large survey of wild-derived mouse isolates for their Prdm9 alleles and relationship to hybrid male sterility or subfertility. They find that full sterility of F1 hybrids occurred in the context of wild genetic backgrounds and fertility status was determined entirely by the Prdm9 genotype. They found that subfertility in some F1s was largely, but not entirely, due to Prdm9 alleles, and was a consequence of defective meiotic chromosome synapsis (presumably due to compromised DSB formation/distribution). Interestingly, they also found that wild-derived t haplotype-bearing mice all had the same Prdm9 allele (t haplotypes, wherein Prdm9 resides, contain inversions that prevent recombination with wild type versions of Chr 17), whereas other alleles were unique to populations in which they were isolated. One important conclusion is that Prdm9 divergence appears to be driven by positive selection at hypervariable sites in the protein.

I found the paper to be carefully written and thorough, but quite dry; it was not easy reading, especially the sections beginning on lines 409 and 505. I wonder if sections of text might be replaced by a better Fig. 1 or a Table.

Overall, the results are straight forward and should be of interest to evolutionary biologists and to Prdm9/recombination aficionados. Again, the major issue for me was the presentation complexity.

As alluded to above, I have 1 suggestion for improving the presentation:

- I don't find Fig. 1 to be very useful....I'm really not sure what I'm looking at here. The legend (which has a glitch reference) says refer to figure S2, but the colored bars don't have obvious meaning to me even after reading the legend. People who have colorblindness will have issues with this. I suggest possibly adding (or replacing) with a Table.

Plus I noticed a couple of mistakes in the figures:

- In Fig. 3, no statistical values or asterisks are shown, just brackets.
- In Fig. 2, rather than using 1-4 asterisks, please just put P value.

Reviewer #3 (Comments for the Authors (Required)):

The manuscript GENETICS-2023-305874 "Natural variation in Prdm9 affecting hybrid sterility phenotypes" describes testing several additional mouse alleles of Prdm9 for their effect on hybrid sterility (expanding previous study by Mukaj et al 2020) and phylogenetic analysis of the

minisatellite repeats in ZFN alleles of Prdm9. In addition, the authors investigated the effect of Hstx2 region haplotypes on hybrid sterility.

While the manuscript is interesting and brings new results, it displays multiple drawbacks. Increased expression of the "h.sterility" dom2 allele of Prdm9 increases the fertility of mouse hybrids [doi: 10.1371/journal.pgen.1003044], but results in no consequences on DOM background. Moreover, decreased Prdm9 expression using a null allele also affects fertility parameters [ibid]. However, the authors have not tested the expression levels of Prdm9 in primary spermatocytes (or prepubertal testes), leaving open the possibility that [some of the] the fertility parameters observed in this study are not due to ZFN but in the expression regulation of Prdm9 (or, albeit less likely, by a hypomorphic mutation in exon/s 1-11). Yet, the authors have not even discussed these alternative explanations.

There is no result on the erosion levels of PRDM9 binding sites by the Prdm9 alleles characterized here and in Mukaj et al 2020, which could support the role of the zinc-finger domain.

There is no attempt to measure or predict DNA sequences binding the characterized ZFNs to find out, which are similar.

I found no statistical evaluation of the hypothesis that the evolutionary age of the zinc-finger domain (with or without introgressed alleles) correlates with fertility parameter/s.

Detailed comments:

1 the species investigated (mouse) should appear in the title

1,35 "Prdm9 diversity" was investigated only in the zinc-finger-encoding exon, please specify 28-44 what was/were the hypothesis/es tested?

28 "Male F1-hybrids" must be more exactly specified, e.g., "These male F1-hybrids"

30 "oligogenic" requires better description

31-33 results on erosion of PRDM9 binding sites are not the subject of this study

31-33 there is no explanation/hypothesis how hybrid sterility relates to asymmetric binding

33-36 "Numerous", "a few", "several" should be replaced by numbers

37 t-haplotypes should be introduced/described

39-40 please specify, how many alleles were found of each type

41 what does it mean "segregated purely"? "co-segregated"?

42-43 I do not see any additional substantiation in this study for "oligogenic control"

54-55 PRDM9 is also expressed in fetal ovaries... reference/s missing

64 references for Prdm9 variability should be moved up, additional references for structural studies supplied

66 wrong references or bad formulation "are predicted"

79 "asymmetric" must be explained in terms of DNA binding by PRDM9

79 unexplained abbreviation "DSBs"

80 unclear writing- where is "ineffective DSB repair" from? How does it stop "forming gametes"?

87 the "cut" does not have to be in the PRDM9-binding motif, the motif erosion is more likely occurring by uneffective repair, not by "cut" itself

94 "initiation asymmetry" should be specified, e.g., "repair initiation asymmetry"
99 what do you precisely mean by "oligogenic" here?
106 "Prdm9 is also a major hybrid sterility gene...": this was in a mix of three subspecies, so better interpretation could be "Prdm9<msc1/dom2> displays also sterilizing effect..."
130 "same Prdm9 allele" should be "same ZNF allele of Prdm9", as no other parts of Prdm9 were analyzed
136 unclear "may be independent"
142-143 info on Prdm9 in t-haplotype occurs twice
160 I do not see any statistical evaluation of the effect of "age of alleles" in this paper
162-165 this sentence lacks clarity, might be too long
168-170 unclear- what variation exactly?

205-236 crucial info missing: the age of the hybrids - because hybrid fertility parameters are known to vary by age [doi: 10.1534/genetics.120.303474, 10.1371/journal.pone.0095806]
if the age is not the same for all males, it has to be given in every plot or in the supplement/Dryad and referred to here [its range stated]
205-236 Were there any differences in phenotyping compared to the previous study (Mukaj et al 2020)?
218-236 How many cells were scored for each male? Is this info deposited in Dryad? If so, its range must be stated here.

337 info redundant with 63,291,477,480,955...
410-412 references missing, initiation unspecified
413-415 need to distinguish divergence in PRDM9-binding domain and PRDM9-binding sites
503-504 unclear what is meant by "clustering of hybrid sterility alleles" - clustering by subspecies of ZFN alleles co-inducing hybrid sterility?
506-507 subject missing?
525-526 Additional explanation need, why was this outcome expected- hybrid sterility is defined as occurring only in offspring.
588 "In reciprocal orientation": do you mean "To test the reciprocal orientation"?
605-607 this requires expanded explanation of the genomic modifiers, as it seems to contradict the previous sentence [603-605]
607-608 This conclusion is only speculative. Prdm9 msc1 and msc6 are not on the same background. Even taking into the account the co-segregation of the Prdm9 alleles with fertility in [outcrossed] hybrids, variance in the Prdm9 besides ZFN exon 12 has not been assessed.
612-615 Again speculation, as the Cst and msc11 ZFN alleles have not been tested in the same system with otherwise 100% genetic background
638 "wild MUS allele": would it be better "wild MUS Prdm9 allele" ?
647 average sperm count must be given instead of "traces of sperm"
647-654 was the transmission ratio distortion checked for these t-haplotype mothers [separately from those having two "fertility" alleles of Prdm9]?
660-661 specify heterogeneous wild genomic

661 specify "heterogeneous wild genomic background"

664 "segregated purely" should read "co-segregated"; this experiment does not exclude other genetically linked variation

685 *msc1* is also present in the PWD strain

711-712 I consider this expected rescue unlikely, because (PWD x B6)F1 hybrids with deleted *dom2* allele (*Prdm9*<*msc14/tm1*>) display very low SC

719 unexplained abbreviation [LD]

727-8 "show reduced fertility": do you mean "show reduced fertility of their hybrids"?

731-34 unclear purpose of this sentence

744-751 Similar statement as in 607-615: I do not see results in this paper showing that these alleles are "on the same background", only from the same population. To exclude background effect/s, resequencing of the LD region containing the ZFN domain would have to be performed in the KH population, as well as expression analysis of *Prdm9*.

751 are you saying SKE and PWD are equal on chromosome 17?

744-751 These statements should be removed or substantiated by sequencing/expression/knock-in experiments.

755-765 is this supported by the frequencies of *Prdm9* alleles in wild mouse populations?

777 "We are grateful to Heike Harre for fertility phenotyping..." This should be either more specific or this person should be among authors, because a substantial part of the paper is about fertility phenotyping.

956 the color code is not explained in Fig.S2 regarding Fig.1, is it in Fig.1 also by chemical properties?

images:

Fig.S1 MMD MMM code is used instead of DOM MUS as stated in Legend

Fig.S5: purple color is very tough to see

Fig.S7-S8: probability stars missing

Minor:

42 etc. "Our data": better "Our results" [data are just numbers, one needs statistics]

76 "PWDxB6": PWD and B6

116 "(B6 X PWD)": "(B6 x PWD)" [cross sign, not a letter]

140, 152, 156 etc. "t" in "t-haplotype/s" should be in italics

651 "P > 0.0001": "P < 0.0001"?

322,324... why are the references italicized?

364 "T/tp⁴" "T" and "tp⁴" should be in italics, "p⁴" in superscript

411 etc. DOM MUS are sometimes in italics, sometimes not - please unify

723-4 "Suppose Individual": "Suppose that individual" ?

Response to Reviewers

Abualia *et al.* “**Natural variation in *Prdm9* affecting hybrid sterility phenotypes**”

Manuscript Number GENETICS-2023-305874

We thank the anonymous peer-reviewers for their detailed comments and thorough review to improve our manuscript. We are excited to read your overall enthusiasm and recognition of our data and appreciate your comments and thorough revision, which has dramatically improved our manuscript's presentation, clarity, and focus.

To the best of our knowledge, we have carefully incorporated all of the referees' suggestions into our manuscript. We have prepared a revised version of the manuscript, which we feel has substantially improved the work, and we hope the extent of revisions meets any remaining concerns.

Below, we provide a detailed, point-by-point response to the specific comments/questions/issues raised in the reviews. The original Reviewer's comments are in *black text, underlined*. Our responses are in blue.

Reviewer #1 (Comments for the Authors (Required)):

The authors look at *Prdm9* alleles in different genetic backgrounds to characterize novel *Prdm9* alleles occurring in natural populations and to assess their effect on fertility.

Overall, this is a nice manuscript. Studies such as these take a long time, and the authors did do experiments carried out in backgrounds with *Prdm9* alleles that haven't been looked at before. However, I am struggling to see what this manuscript teaches us, that hasn't been shown before.

We are grateful to the Reviewer for their positive comments on our paper, and we have addressed their queries in detail below.

First, compared to the work by Mukaj *et al.* 2020, the current work extends our understanding of *Prdm9*-mediated hybrid sterility outside of the context of inbred lines of mice. Our data expand the occurrence of subfertility in some F₁ hybrids by Mukaj *et al.* 2020, and show that it is a consequence of defective meiotic chromosome synapsis (presumably due to compromised DSB formation/distribution) in mice despite a high level of background heterogeneity.

We learned that:

- The full sterility of F1 hybrids can occur in the context of wild genetic backgrounds (and not in inbred lines) and that the fertility status was under the control of the *Prdm9* genotype.
- Despite extremely high polymorphism of *Prdm9* in natural populations, all *t*-haplotype-bearing mice shared the same *Prdm9* allele
Based on a novel phylogenetic analysis of the minisatellite repeats in ZFN alleles of *Prdm9* all currently identified hybrid sterility *Prdm9* alleles are subspecies-specific with considerable divergence from a common ancestor. .

In the revised version of the manuscript, we added DNA-binding predictions of the characterized ZNFs and tested, for the first time, the overlap in predicted binding sites of the wild alleles.

I have a number of comments.

Line 202 of methods, a character is missing.

We have added the missing character

Line 260, "to ensure d high-fidelity", "d" is an error

We have removed the "d"

Line 273: "Only PRDM9 variants with less than 12 ZNFs..." How many alleles are expected to be larger than 12 ZNFs? How many alleles are lost due to this constraint?

-
We thank the Reviewer for this suggestion. We have now clarified that the majority of alleles are larger than 12 ZNF but that no alleles were lost due to this constraint as all larger alleles were nevertheless fully sequenced, just forward and reverse sequences did not overlap along their entire length. We have rephrased the sentence to read the following:

Exon 12 of *Prdm9* was fully sequenced for all alleles; however, forward and reverse sequences overlapped along the entire length of the exon only in alleles smaller than <1000 bp, such that larger alleles had sequence stretches only covered by either forward or reverse sequencing. Nevertheless, the sequencing products of all alleles still provided sufficient overlap for full-length assembly.

Line 345-6: "... human *Prdm9* allele B, able to rescue sterility when genetically engineered into mice" This is meant to say that the *Prdm9* allele B rescues sterility, not that the ancestral Glycine alone can rescue?

We thank the Reviewer for this suggestion. The effect of the ancestral glycine is not implied as the deciding factor in this rescue. We have rephrased the sentence to read the following:

The nucleotides coding for Glycine in this position appear to be the ancestral alleles, as the same amino acid is also encoded at this position in human PRDM9, included in the genetically engineered "humanized" allele B in mice, where it can rescue sterility (Davies et al. 2016)(**Error! Reference source not found.**).

Line 374: to clarify, since the t-haplotype frequency took over the mouse populations while housed in the lab, the *Prdm9* alleles identified do not reflect the PRDM9 variants in wild populations?

Thank you for this comment. We have rephrased the paragraph to clarify that we only meant to imply that allelic frequencies have likely changed due to the transmission ratio distortion.

Even though the alleles we identified in the outcrossed population reflect the *Prdm9* alleles present at the catching locations in the wild, their initial population frequencies have gotten heavily distorted due to the *t*-haplotype over-transmission, the alleles-frequencies may not reflect the initial population frequencies in which they occurred in the wild.

Line 376-393: Can you speak to how these amino acid changes are predicted to affect DNA binding specificity?

We agree with the Reviewer that the crucial point on PRDM9 binding was not explained well enough. To clarify, we have now included cartoon depicting how DNA binding is determined by amino-acids in positions -1, 3, and 6 of the alpha-helix of each C₂H₂ ZNFs in Figure 3A.

Line 557: how did you determine spermatocytes were in pachytene, and not late zygotene or early diplotene

Thank you for these important questions. Indeed, staging can be quite subjective to the untrained eye, especially when relying only on a single marker, such as sycp3, alone. However, we also used γH2AX staining that localizes to transcriptionally silenced chromatin, and have added the following explanation:

To determine if cells are in the pachytene stage, when the synaptonemal complex is fully formed, we immunostained synaptonemal-complex protein 3 (SYCP3). In contrast to continuous SYCP3 staining in pachytene, SYCP3 staining is disorganized and patchy in the preceding late zygotene stage (when the synaptonemal complex is still forming) and the succeeding early diplotene stage (when the synaptonemal complex begins to disassemble as autosomes de-synapse and lateral elements separate), when the SYCP3 signal becoming visible as pair of thinner threads (DE LA FUENTE *et al.* 2007). Additionally, we assessed H2AX, which localizes to only the X and Y chromosome in males only at mid-to late Pachytene, forming a punctate “sex body” of transcriptional silencing (FERNANDEZ-CAPETILLO *et al.* 2003).

Line 651-654: since t-haplotype is known to have transmission ratio distortion in males, isn't it impossible to say that your observations are due to *Prdm9* fertility modifiers? Can you say this for *Prdm9* alleles that are not associated with t-haplotype? And if so, wouldn't it be better to make that argument with these alleles.

Thank you for pointing out, that this part of the discussion was not clearly written. We have rephrased the entire paragraph to read the following:

The apparent trans-effect of *Prdm9*, located on the *t*-haplotype, cannot be disentangled from that of other fertility modifiers within a single haplotype block. However, while testis *Prdm9* expression levels in mice with *t*-haplotypes were not significantly different from those in mice without *t*-haplotypes, several other genes on *t*-haplotypes are enriched for copy gain events (KELEMEN AND VICOSO 2018) and show overexpression in the testis of *t^{wt}* heterozygous mice (KELEMEN *et al.* 2022).

Figure 1: Can you add the number of ZNF repeats?

Thank you very much for this valuable comment. We have completely remade Figure 1 (which is now Figure 3) also in response to comments made by other reviewers. The new Figure now contains a cartoon overview of the entire ZNF domain, in a tabular structure with the number of each ZNF on top.

Explain what the colors of each dot next to the allele means. For example, MGI:5503021 *Mmd* dom4 SCHEST has a red color. What does red mean?

Thank you for asking for clarification, we have now improved the figure legend. Indeed, the red color denotes DOM (*Mus musculus domesticus*), blue color denotes MUS (*Mus musculus*)

musculus) and purple color denotes the t-haplotype allele *mmt1* found in both subspecies. We have added this explanation also to the Figure legends to clarify.

Somewhere, can you speak to the amount of variation between the mouse strains you are looking at. Like, DOM strains contain X number of diverged nucleotides per 1KB, or something, to get an idea of how, beyond phylogeny, diverged the genomes are?

Thank you for this valuable comment. As well as spending more effort in outlining that these are not strains of mice, but instead an outcrossed population of mice with inter-individual genetic diversity. We have added extra detail on the amount of variation in our mouse dataset, included a new paragraph and added Supplementary Table S1 with average amounts of genomic variation, compared to the C57Bl6J/strain.

Table S1: Variation, Introgression, and Recombination rate of outcrossed populations with data from (a) (LAWAL *et al.* 2021) (b) (WOOLDRIDGE AND DUMONT 2022) (c) (BANKER *et al.* 2022) (d) (STAUBACH *et al.* 2012)

outcrossed populations	# SNPS	SNP density (bp/SNP)	# population private variants	Average introgression per individual (%)	% of the genome affected by introgression
AKH	10,937,288 ^a	225 ^b	1,872,782 ^a	NA	4.90 ^d
MCF	11,108,085 ^a	222 ^b	2,208,483 ^a	0.069 ^c	2.80 ^d
CBG	11,930,888 ^a	206 ^b	545,881 ^a	0.068 ^c	2.80 ^d
AHI	17,877,283 ^a	138 ^b	3,333,440 ^a	0.036 ^c	NA

Line 956: some error in this line of text

Thank you for your comment. We have rephrased this paragraph to correct the error.

(A) Sperm count, and **(B)** paired testes weights for intersubspecific offspring of (B6.DX1s × wild MUS) or (PWD × wild DOM) crosses grouped by *Prdm9* genotype and compared to offspring of known hybrid sterility crosses (B6.DX1s × PWD) and (PWD × B6). The asterisks refer to the significance values of pairwise ANOVA with Bonferroni correction, with $P < 0.0332$ (*), $P < 0.0021$ (**), $P < 0.0002$ (***), and $P < 0.0001$ (****). All hybrid males carry the *Hstx1^{PWD}* allele on Chr X. **(C)** The panels show spermatocyte spreads of two intersubspecific B6.DX1s × wild MUS hybrids, with differing *Prdm9* genotypes. To identify spermatocytes at the pachytene stage, antibody staining against Synaptonemal complex Protein SYCP3 (red) and γH2AX (grey), which mark chromatin associated with asynapsed axes at the zygotene/pachytene transition. DNA was counterstained with DAPI (blue) and localized grey dots represent CEN-labeled centromeres. Defects in chromosome asynapsis were assessed by antibody staining for HORMAD2 protein (green), which marks asynapsed autosomal chromosomes in addition to the nonhomologous parts of XY sex chromosomes that are physiologically observed in normally progressing meiocytes. **(D)** The percentage of asynaptic cells on the Y-axis, were grouped by *Prdm9* genotype. The percentage of asynaptic cells correlated with fertility parameters of intersubspecific F₁-hybrids, namely **(E)** sperm count and **(F)** paired testes weights, with red dotted lines representing 95% confidence intervals, and **P** values, and Pearson values **r**, given on the bottom right.

I've noticed that the testis weights do not always correlate to sperm counts. This either means, testis weight is a bad parameter for fertility, or I suspect more likely, you did not control for body weight. If you do average testis weight normalized to animal body weight, I expect that

testis weight will correlate with sperm counts/fertility much better. Otherwise, id reconsider using testis weight and look at sperm counts only

Thank you for this valuable comment. In our outcrossed mice, the testis weight to sperm count correlations were stronger than testis weight normalized to animal body weight, presumably because we see larger fluctuations in body weight, as well as large differences in lean body mass between outcrossed mouse individuals. We have now included sperm counts correlated with averaged testis weight normalized to animal body weight into Supplementary Figure S7, which now contains all correlation analyses of fertility parameters, and in all types of performed crosses.

Figure S1: Correlation analyses of fertility parameters in all types of performed crosses. The type of crossing scheme is (A) Intraspecific cross and different (B) Interspecific, with we tested the relationship of paired testes weight to body weight, as well as the sperm count to TW/BW ratio, and testes weights (from left to right),

Figure S2/3: what does the yellow shade and pink shade mean?

Thank you very much for your comment. As the zinc-finger domain of PRDM9 is comprised of Cysteine-2-Histidine-2 (C₂H₂) type ZNFs we highlight the Cysteines and Histidine residues in yellow. We used the yellow shade to highlight all amino acid positions that differ from the majority, which included the DNA-binding ZNF residues in position -1, 3 and 6 of the alpha-helix, that are responsible for DNA binding specificity. We have rephrased the legend accordingly, which now reads:

Neighbor-joining tree calculated on the nucleotide sequences of all PRDM9 alleles in this study and (Mukaj *et al.* 2020), that code for the C₂H₂ ZNFs array, with red nodes for DOM and blue nodes for MUS alleles, and with purple nodes depicting the t-haplotype allele found in MUS and DOM

Line 1010: there is some error in this line of text

Thank you for the We have rephrased the text as follows:

(A) MUS from Kazakhstan without t-haplotypes, all offspring are sorted by *Prdm9* genotype, with the father (sire) ID on the X-axis. One sire (50054290) was homozygous for *Prdm9*, all others heterozygous for different *Prdm9* alleles **(B)** Offspring with t-haplotypes, grouped by *source* population of the sires, and *sire IDs* on the X-axis. (left) MUS from Kazakhstan (right) DOM from Cologne-Bonn, Germany (CBG), Massif-Central, France (MCF) and Ahvaz, Iran (AHI).

Reviewer #2 (Comments for the Authors (Required)):

This paper describes a large survey of wild-derived mouse isolates for their *Prdm9* alleles and relationship to hybrid male sterility or subfertility. They find that full sterility of F1 hybrids occurred in the context of wild genetic backgrounds and fertility status was determined entirely by the *Prdm9* genotype. They found that subfertility in some F1s was largely, but not entirely, due to *Prdm9* alleles, and was a consequence of defective meiotic chromosome synapsis (presumably due to compromised DSB formation/distribution). Interestingly, they also found that wild-derived t haplotype-bearing mice all had the same *Prdm9* allele (t haplotypes, wherein *Prdm9* resides, contain inversions that prevent recombination with wild type versions of Chr 17), whereas other alleles were unique to populations in which they were isolated. One important conclusion is that *Prdm9* divergence appears to be driven by positive selection at hypervariable sites in the protein.

I found the paper to be carefully written and thorough, but quite dry; it was not easy reading, especially the sections beginning on lines 409 and 505. I wonder if sections of text might be replaced by a better Fig. 1 or a Table. Overall, the results are straight forward and should be of interest to evolutionary biologists and to *Prdm9*/recombination aficionados. Again, the major issue for me was the presentation complexity.

As alluded to above, I have 1 suggestion for improving the presentation:

- I don't find Fig. 1 to be very useful. I'm really not sure what I'm looking at here. The legend (which has a glitch reference) says refer to figure S2, but the colored bars don't have obvious meaning to me even after reading the legend. People who have colorblindness will have issues with this. I suggest possibly adding (or replacing) with a Table.

We agree with the Reviewer and have removed the original Figure 1, have improved the presentation and added a cartoon to depict that the colored boxes represent those variable amino acids positions, which are responsible for DNA binding specificity of PRDM9. We corrected the glitch reference, and ensured that the colors would be accessible to individuals with the most common types of colorblindness's using the program "Color Oracle"(VIENOT *et*

a/. 1999) which simulates how individuals with different color vision deficiencies perceive colored images.

Plus, I noticed a couple of mistakes in the figures:

- In Fig. 3, no statistical values or asterisks are shown, just brackets.

Thank you very much for this comment. We have now added the statistical values to the figure.

- In Fig. 2, rather than using 1-4 asterisks, please just put P value.

We have now added P-values instead of using asterisks.

Reviewer #3 (Comments for the Authors (Required))

The manuscript GENETICS-2023-305874 "Natural variation in Prdm9 affecting hybrid sterility phenotypes" describes testing several additional mouse alleles of *Prdm9* for their effect on hybrid sterility (expanding previous study by Mukaj et al 2020) and phylogenetic analysis of the minisatellite repeats in ZFN alleles of *Prdm9*. In addition, the authors investigated the effect of Hstx2 region haplotypes on hybrid sterility.

While the manuscript is interesting and brings new results, it displays multiple drawbacks. Increased expression of the "h.sterility" dom2 allele of Prdm9 increases the fertility of mouse hybrids [doi: 10.1371/journal.pgen.1003044], but results in no consequences on DOM background. Moreover, decreased Prdm9 expression using a null allele also affects fertility parameters [ibid]. However, the authors have not tested the expression levels of Prdm9 in primary spermatocytes (or prepubertal testes), leaving open the possibility that [some of the] the fertility parameters observed in this study are not due to ZFN but in the expression regulation of *Prdm9* (or, albeit less likely, by a hypomorphic mutation in exon/s 1-11). Yet, the authors have not even discussed these alternative explanations.

Thank you for these comments. In the aforementioned study the levels of *Prdm9* expression at the mRNA level in prepubertal testis were similar in sterile and fertile hybrids, and only decreased when the *Prdm9*^{dom2} allele was removed. Moreover, decreased level in (B6xPWD)Prdm9^{-msc1} males improved, not decreased the fertility traits, just the opposite of what would be expected, if expression level decided the fertility status. The effect of extra *Prdm9* copy on fertility of hybrids was thoroughly discussed in (doi:10.1038/nature16931) and fits better to the *Prdm9* binding site asymmetry hypothesis than to the effect of expression regulation. Furthermore, reduced expression of *Prdm9* mRNA in testis is not detectable in testis mRNA expression data of the outcrossed populations. This data was generated by (Harr et al. 2016) and is publicly available, showing robust *Prdm9* testis expression across many different individuals of each population (KELEMEN et al. 2022).

There is no result on the erosion levels of PRDM9 binding sites by the *Prdm9* alleles characterized here and in Mukaj et al 2020, which could support the role of the zinc-finger domain. There is no attempt to measure or predict DNA sequences binding the characterized ZFNs to find out, which are similar.

Thank you for this valuable comment. We have now added the prediction of DNA sequences bound by the characteristic ZNFs, and added an analysis to find out which of the DNA binding motifs are similar, including analyses on the genomic overlap of predicted DNA binding motifs.

I found no statistical evaluation of the hypothesis that the evolutionary age of the zinc-finger domain (with or without introgressed alleles) correlates with fertility parameter/s.

We apologize for being unclear in our explanations, as we had never meant to imply any correlation of an evolutionary age of alleles with any of fertility parameters, and have taken care to clarify that the nucleotide hamming distances showed a surprising pattern that the alleles which were known to induce hybrid sterility, were not among the oldest alleles as we had hypothesized.

Detailed comments:

Title:

1, the species investigated (mouse) should appear in the title

1,35 "Prdm9 diversity" was investigated only in the zinc-finger-encoding exon, please specify

We thank the Reviewer for this comment. As the Reviewer correctly notes, the allelic diversity pertains to exon12 of Prdm9.

We have amended the title to now read:

Natural variation in the zinc-finger-encoding exon of Prdm9 affects hybrid sterility phenotypes in mice.

And amended the abstract as follows.

We identified novel Prdm9 Exon 12 alleles, encoding the DNA-binding domain of the PRDM9 protein in outcrossed wild mouse populations from Europe, Asia, and the Middle East.

28-44 what was/were the hypothesis/es tested?

We thank the Reviewer for this comment and have added a statement outlining our hypotheses to the end of the introduction section.

In summary, the low incidence of sterile wild-mouse hybrids in DOM/MUS natural hybrid zone (citation), together with the reported large number of hybrid sterility loci in intersubspecific backcrosses and intercrosses, contrasts with the F₁ hybrid sterility model based on the Prdm9 allelic incompatibility, Hstx2, and background heterozygosity in PRDM9 binding sites. Further experimental evidence is needed to understand the mechanism of Prdm9-driven hybrid sterility and its role in wild mouse populations. Here, we ask whether hybrid sterility is under the PRDM9 control in wild mice beyond the context of inbred strains of mice. If the Prdm9-driven hybrid sterility is linked to the asymmetric erosion of PRDM9 binding motifs, then in a simple scenario, the sterility-inducing alleles would be expected to be ancestral alleles situated closest to the common ancestor on the phylogenetic tree. To test this hypothesis we examine the evolutionary relationship between the known Prdm9 hybrid sterility alleles and any newly identified allele.

28 "Male F1-hybrids" must be more exactly specified, e.g., "These male F1-hybrids"

We thank the Reviewer for this comment and have rephrased the sentence accordingly.

These male F₁-hybrids fail to complete chromosome synapsis and arrest meiosis at prophase I, due to incompatibilities between the Prdm9 gene and hybrid sterility locus Hstx2.

30 "oligogenic" requires better description

Thank you very much for this comment. Due to word restrictions, and this phrase has been removed from the abstract, and is explained in more detail in the introduction now:

In this cross, fertility was restored when the B6 PRDM9 zinc-finger array was replaced with the human variant B, making symmetric recombination hotspots predominant (Davies *et al.* 2016), further supporting the hypothesis that hybrid sterility is under an oligogenic control, with PRDM9 as the main factor.

31-33 results on erosion of PRDM9 binding sites are not the subject of this study

Thank you for this valuable comment. We have substantially rephrased several paragraphs of the manuscript to explain better how we infer the level of erosion from the fertility data and published results, and hope that we have now clarified our writing sufficiently.

In the Abstract we now state:

..., the combination of results from intra- and intersubspecific crosses suggest that the same pattern of erased PRDM9 *msc1* binding motif may be shared by MUS mice with different *Prdm9* alleles, implicating uncoupling of erased PRDM9 binding motifs from presence of PRDM9 zinc finger arrays at the population level.

Based on our experiments which we describe starting at line 517:

According to the hypothesis linking the *Prdm9*-driven hybrid sterility to the asymmetric erosion of PRDM9 binding motifs, the sterility of F₁ hybrids results from the erosion of MUS PRDM9^{*msc1*} binding sites in the PWD genome and DOM PRDM9^{*dom2*} sites in the B6 genome (Davis et al. 2016). If a given MUS population with an unrelated “fertility” *Prdm9* allele carries the same *msc1* binding motif as PWD mice, then the fertility of (PWD x MUS) x B6 hybrids would segregate according to *Prdm9*; *Prdm9*^{*msc1*} males would be sterile, and their *Prdm9*^{*MUS*} siblings would be fertile. However, if the given wild MUS genome does not carry any PRDM9 binding site erasure or the erased motifs match an unrelated *Prdm9* allele, then *Prdm9*^{*msc1*} and *Prdm9*^{*MUS*} male progeny should be fertile. The same assumption can be made for the outcome of PWD x (B6 x DOM) testcross to test the coupling of eroded PRDM9 binding motif with corresponding PRDM9 allelic zinc finger domains. To test the decoupling of eroded PRDM9 binding sites from the corresponding allelic form of *Prdm9* in wild populations, we generated intraspecific (DOM x B6) hybrid males and crossed them to PWD females to test the DOM-derived *mmt1* and *dom2* alleles against the mixed DOM background. To test MUS alleles and their genetic background, the B6.DX1s females were crossed with intraspecific (PWD x MUS) F₁ hybrid males. We also performed an analogous cross in reciprocal orientation for MUS alleles, where (PWD x wild MUS) F₁ hybrid females were crossed with B6 males.

And the discussion of our results starting at line 704:

While the data on *Prdm9* polymorphism in the wild house mouse populations are accumulating, and indeed the *Prdm9* genes show remarkable natural allelic divergence, with more than 150 alleles having been found in mouse populations to date (BUARD et al. 2014; KONO et al. 2014; VARA et al. 2019; MUKAJ et al. 2020), little is known about their DNA binding motifs and their degree of erosion. In this context, the finding that five populations with fertile *Prdm9* alleles carried evidence of eroded *msc1* hotspots was surprising, suggesting a decoupling of the evolutionary dynamics of the PRDM9 zinc-finger domains and their binding sites. In other words, the erosion of the *msc1* hotspots may be much more common in natural populations than the *msc1* allele itself. The observation that recombination maps based on linkage disequilibrium (LD) analyses revealed significant overlap of historical hotspots of the AKH population with contemporary *msc1* activated hotspots, detected using DMC1 Chromatin Immunoprecipitation and sequencing (ChIP-Seq) in the PWD strain. At the same time only weak conservation of recombination maps as each population (AHI, MCF, CGB and AKH,) had mostly unique hotspots (WOOLDRIDGE AND DUMONT 2022).

31-33 there is no explanation/hypothesis how hybrid sterility relates to asymmetric binding

This phrase has been removed from the abstract, and is explained in more detail in the introduction, which now reads:

In mouse hybrids, higher affinity binding sites for a given PRDM9 variant are four times more likely to be found on the chromosome of the other species, with which *Prdm9* did not coevolve, suggesting that binding site erosion is a predominant factor driving hotspots loss in several mouse lineages (SMAGULOVA *et al.* 2016). Indeed, in sterile hybrids of inbred strains PWD (MUS) x B6 (DOM), a bias in initiation efficiency between diverged homologs is mainly driven by functional *dom2* binding sites found on the PWD genome that are eroded on the B6 genome and vice versa for *msc1* sites (DAVIES *et al.* 2016), with recombination being initiated at a large number of asymmetric sets of breaks (MIHOLA *et al.* 2009; DAVIES *et al.* 2016). In this cross, fertility was restored when the B6 PRDM9 zinc-finger array was replaced with the human variant B, making symmetric recombination hotspots predominant (DAVIES *et al.* 2016), further supporting the hypothesis that hybrid sterility is under an oligogenic control, with PRDM9 as the main factor.

33-36 "Numerous", "a few", "several" should be replaced by numbers

We apologies that we did not specify the numbers and have now remedied this:

We tested whether *Prdm9*-mediated reproductive isolation operates in populations of mice with greater genetic diversity than any wild-derived inbred strain used to date, for seven MUS *Prdm9* alleles in MUS populations, as well as the *t*-haplotype allele in one MUS and three DOM populations.

37 t-haplotypes should be introduced/described

Thank you for this valuable comment. We have added an introduction of *t*-haplotypes in the Abstract, as follows:

The same *Prdm9* allele was found in all mice bearing *t*-haplotypes, an introgressed Chr17 haplotype encompassing *Prdm9* and inversions preventing recombination with wild type Chr17.

t-haplotypes are also further described in the introduction section:

The *t*-haplotype consists of 30 Mb of introgressed sequence transferred from an unidentified *Mus* ancestor in the *Mus musculus* subspecies over one million years ago (HAMMER AND SILVER 1993) and encompasses the *Prdm9* locus (TRACHTULEC *et al.* 2008), with the most diverse allele identified to date in several mice of the *Mus musculus* subspecies (KONO *et al.* 2014).

39-40 please specify, how many alleles were found of each type

Thank you for this comment, we have now specified how many alleles of each type were found.

We tested whether *Prdm9*-mediated reproductive isolation operates in populations of mice with greater genetic diversity than any wild-derived inbred strain used to date, for seven MUS *Prdm9* alleles in MUS populations, as well as the *t*-haplotype allele in one MUS and three DOM populations.

41 what does it mean "segregated purely"? "co-segregated"?

Thank you for this comment, we have removed this statement from the abstract, and have amended our statements in the results section to incorporate your helpful suggestion:

As previously observed with the *Hstx2^{PWD}* sterility allele, fertility co-segregated predominantly with the *Prdm9* genotype, irrespective of which X-chromosomal haplotype the offspring possessed.

42-43 I do not see any additional substantiation in this study for "oligogenic control"

We thank you for this comment. We have now removed the statement from the abstract

54-55 PRDM9 is also expressed in fetal ovaries... reference/s missing.

We agree with the Reviewer and have added that PRMD9 is also expressed in fetal ovaries and have included the appropriate references for fetal ovary expression.

The PRDM9 protein is expressed in testicular tissue and fetal mouse ovaries during the early phases of meiotic prophase I when recombination is initiated (HAYASHI *et al.* 2005; LAWSON *et al.* 2011).

64 references for *Prdm9* variability should be moved up, additional references for structural studies supplied

We thank the Reviewer for this suggestion, and apologize that we did not include sufficient references. We have rephrased the passages, moved up the references for *Prdm9* variability and supplied additional references for structural studies.

PRDM9 has three conserved domains, an N-terminal KRAB domain that promotes protein-protein binding (IMAI *et al.* 2017; PARVANOV *et al.* 2017; WANG *et al.* 2021), an NLS/SSXR domain with nuclear localization signal, and a central PR/SET domain that confers methyltransferase activity (POWERS *et al.* 2016). The C-terminal domain is highly polymorphic and comprises an array of C₂H₂-type zinc fingers (ZNFs) which differ among PRDM9 variants in both type and number (OLIVER *et al.* 2009; BAUDAT *et al.* 2010; BERG *et al.* 2010; PARVANOV *et al.* 2010; BERG *et al.* 2011; BAUDAT *et al.* 2013; BUARD *et al.* 2014; KONO *et al.* 2014).

66 wrong references or bad formulation "are predicted"

We thank the Reviewer for this suggestion and have rewritten the sentence to clarify.

Additional amino acid substitutions in positions -5, -2, and 1 of the α -Helix are rarely seen (Parvanov *et al.* 2010; Kono *et al.* 2014). Since only amino acids at positions -1, 3, and 6 are involved in protein-DNA interactions all other positions are not predicted to affect DNA binding affinity (Persikov and Singh 2014).

79 "asymmetric" must be explained in terms of DNA binding by PRDM9

and

79 unexplained abbreviation "DSBs"

and

80 unclear writing- where is "ineffective DSB repair" from? How does it stop "forming gametes"?

We apologies that these passages were not clear, and have thoroughly rephrased this section and explained the abbreviation.

Defective pairing and high levels of chromosomal asynapsis are observed in hybrids with ineffective double-stranded break (DSB) repair (MIHOLA *et al.* 2009; DAVIES *et al.* 2016). It has

been hypothesized that the molecular mechanism of PRDM9 action is related to the evolutionary divergence of homologous genomic sequences in DOM and MUS subspecies (DAVIES *et al.* 2016) and, more specifically, to the phenomenon of historical erosion of genomic binding sites of PRDM9 ZNF domains (DAVIES *et al.* 2016; FOREJT 2016; ZELAZOWSKI AND COLE 2016; FOREJT *et al.* 2021). Nucleotide polymorphisms within genomic target motifs may affect the binding affinity of PRDM9 in heterozygous individuals. The preferential formation of DSBs occurs on the haplotype with the motif with a stronger binding affinity (BAKER *et al.* 2015). However, since the uncut strand provides the template for repair, the less efficient motif is preferentially transmitted to the next generation, which can lead to the erosion of PRDM9 binding sites over time (JEFFREYS AND NEUMANN 2002). Therefore, polymorphisms that reduce the binding affinity of a given PRDM9 variant are predicted to become enriched within populations over time, resulting in the attenuation of the hotspots (BOULTON *et al.* 1997). Direct evidence for over-transmission has been observed in human and mouse hotspots (JEFFREYS AND NEUMANN 2005; BERG *et al.* 2011; COLE *et al.* 2014; ODENTHAL-HESSE *et al.* 2014). In mouse hybrids, higher affinity binding sites for a given PRDM9 variant are four times more likely to be found on the chromosome of the other species, with which *Prdm9* did not coevolve, suggesting that binding site erosion is a predominant factor driving hotspots loss in several mouse lineages (SMAGULOVA *et al.* 2016).

87 the "cut" does not have to be in the PRDM9-binding motif, the motif erosion is more likely occurring by ineffective repair, not by "cut" itself

and

94 "initiation asymmetry" should be specified, e.g., "repair initiation asymmetry"

We apologize that this was not clear, and have rephrased the offending sentence accordingly.

Indeed, in sterile hybrids of inbred strains PWD (MUS) x B6 (DOM), a bias in initiation efficiency between diverged homologs is mainly driven by functional *dom2* binding sites found on the PWD genome that are eroded on the B6 genome and vice versa for *msc1* sites (DAVIES *et al.* 2016), with recombination being initiated at a large number of asymmetric sets of breaks (MIHOLA *et al.* 2009; DAVIES *et al.* 2016)

99 what do you precisely mean by "oligenic" here?

We apologize for the typo, indeed we meant "Oligogenic" not oligenic. However, this paragraph has been rephrased also in response to previous comments. We hope it is now clear what we meant.

In this cross, fertility was restored when the B6 PRDM9 zinc-finger array was replaced with the human variant B, making symmetric recombination hotspots predominant (Davies *et al.* 2016), further supporting the hypothesis that hybrid sterility is under an oligogenic control, with PRDM9 as the main factor.

106 "*Prdm9* is also a major hybrid sterility gene...": this was in a mix of three subspecies, so better interpretation could be "*Prdm9*^{msc1/dom2} displays also sterilizing effect..."

We agree entirely with the Reviewer that *Prdm9* ^{dom2/msc1} had a sterilizing effect also in a mix of three subspecies, and have rephrased our statement accordingly.

Prdm9^{msc1/dom2} also displayed a sterilizing effect in MUS x CAS hybrids, where the rate of synapsis was proportional to the level of non-recombining MUS genetic background (VALISKOVA *et al.* 2022).

130 "same *Prdm9* allele" should be "same ZNF allele of *Prdm9*", as no other parts of *Prdm9* were analyzed

Thank you for this valuable comment, as you are indeed correct, that the ZNF-alleles of *Prdm9* are the focus of this study. We have now taken great care to clarify that the effect of PRDM9-mediated sterility was evaluated based on variability in the ZNF-encoding nucleotide sequence, and have changed not only the title, but are now only referring to the ZNF alleles of *Prdm9* throughout, using phrases such as:

"*Prdm9* allele coding for the zinc-finger domain ..." or

"ZNF domain encoding alleles" or

"...alleles of *Prdm9* in Exon 12, encoding the DNA-binding domain of the PRDM9 protein"

We have also added an explanation as to why we are most interested in the ZNF domain encoding alleles to the introduction, which now reads:

In this cross, fertility was restored when the B6 PRDM9 zinc-finger array was replaced with the human variant B, making symmetric recombination hotspots predominant (Davies *et al.* 2016), further supporting the hypothesis that hybrid sterility is under an oligogenic control, with PRDM9 as the main factor.

136 unclear "may be independent"

We apologize that it was not clear what we meant. We have rephrased our statement to clarify.

Several interchangeable autosomal loci have been proposed that may suffice to activate the Dobzhansky-Muller incompatibility in wild mouse hybrids (DZUR-GEJDOSOVA *et al.* 2012; TURNER AND HARR 2014), but it has not been tested whether these loci interact with *Prdm9*.

142-143 info on *Prdm9* in t-haplotype occurs twice

We apologize for the apparent redundancy. In the first sentence we refer to studies that showed that the *Prdm9* locus is present on the t-haplotype. The second reference pertains to an observation that a particularly diverse *Prdm9* allele was seen to be associated with t-haplotypes in wild mice. We have now clarified this sentence:

The t-haplotype consists of 30 Mb of introgressed sequence transferred from an unidentified *Mus* ancestor in the *Mus musculus* subspecies over one million years ago (HAMMER AND SILVER 1993) and encompasses the *Prdm9* locus (TRACHTULEC *et al.* 2008), with the most diverse allele of the *Mus musculus* subspecies identified to date (KONO *et al.* 2014).

160 I do not see any statistical evaluation of the effect of "age of alleles" in this paper.

We apologize for the confusion, as we had not meant to imply to look at allelic age, but instead to the context of evolutionary divergence times between alleles using phylogenetic reconstructions of the *Prdm9* hypervariable minisatellite, that coded for the ZNF array responsible for the DNA binding affinity of the PRDM9 protein.

The sentence in question has since been removed, however, we now added another paragraph to detail what we meant.

The phylogenetic analyses further support a scenario in which MUS and DOM have split recently and are still speciating but do not reveal any apparent clustering of alleles co-inducing hybrid sterility by subspecies. Admittedly, no evidence was found to support the idea that the hybrid sterility susceptible alleles (*msc1*, *msc1*, *msc5*, *dom2*, *dom3*, and *dom5*) belong to the evolutionary oldest ones closest to the common ancestor. On the contrary, the *msc1*, *msc2*, and *dom3* alleles are the most distal, sitting on the most distant branch of the phylogenetic tree (**Error! Reference source not found.** and **Error! Reference source not found.**).

162-165 this sentence lacks clarity, might be too long

We apologize for being unclear and have rephrased the sentence to clarify what we mean. To improve clarity, we also split the sentence into two.

We investigated whether seven novel *Prdm9* alleles in MUS populations, as well as the *t*-haplotype allele in one MUS and three DOM populations, induced *Prdm9*-mediated reproductive isolation.

168-170 unclear- what variation exactly?

We are grateful for this feedback, as it allowed us to elude to the genomic variation in more detail in the Results section. (starting in line 364). We has, also added Supplementary table S1, to summarize the amount of genetic diversity compared to the C57BL/6 strain.

Table S2: Variation, Introgression, and Recombination rate of outcrossed populations with data from (a) (LAWAL *et al.* 2021) (b) (WOOLDRIDGE AND DUMONT 2022) (c) (BANKER *et al.* 2022) (d) (STAUBACH *et al.* 2012)

outcrossed populations	# SNPS	SNP density (bp/SNP)	# population private variants	Average introgression per individual (%)	% of the genome affected by introgression
AKH	10,937,288 ^a	225 ^b	1,872,782 ^a	NA	4.90 ^d
MCF	11,108,085 ^a	222 ^b	2,208,483 ^a	0.069 ^c	2.80 ^d
CBG	11,930,888 ^a	206 ^b	545,881 ^a	0.068 ^c	2.80 ^d
AHI	17,877,283 ^a	138 ^b	3,333,440 ^a	0.036 ^c	NA

In addition, we rephrased the sentence to clarify that these genetically diverse individuals provide a unique resource for testing the oligogenic *Prdm9*-mediated outside of the context of inbred strains of mice.

These populations have been housed and maintained as outcrosses for many generations before this study (see Materials and Methods). Despite high degrees of relatedness, these populations have maintained a much larger genomic diversity than inbred strains (LAWAL *et al.* 2021). These outcrossed populations show low levels of introgression between MUS/DOM (Ullrich *et al.* 2017) and moderate levels of bidirectional introgression patterns from *Mus spretus* in all three DOM populations (Banker *et al.* 2022). These outcrossed populations of mice with inter-individual genetic diversity, have high average SNP densities compared to the C57BL/6 strain, with the number of population-private variants, and genomic introgression for each population collected in (**Table S2**). The distribution of original trapping locations of all mice tested for hybrid sterility phenotypes in this study and in (MUKAJ *et al.* 2020) are shown in **Error! Reference source not found.**. Testis mRNA expression levels are available for multiple individuals of each outcrossed population (Harr *et al.* 2016) and have demonstrated a robust *Prdm9* expression (KELEMEN *et al.* 2022). In summary, these genetically diverse individuals

from several outcrossed populations provide a unique resource to evaluate the *Prdm9* allelic incompatibility-mediated hybrid sterility model beyond inbred strains of mice.

205-236 crucial info missing: the age of the hybrids - because hybrid fertility parameters are known to vary by age [doi: 10.1534/genetics.120.303474, 10.1371/journal.pone.0095806] if the age is not the same for all males, it has to be given in every plot or in the supplement/Dryad and referred to here [its range stated]

We agree fully with the Reviewer. To clarify that we were aware of the age-based variation, and although the age had already been included in the Dryad repository, and we had already restricted our analyses to an age range where the parameters are relatively stable, we have now added a supplementary Figure (Figure S9) and the following statement:

We analyzed the fertility parameter for all mice aged 60-100 (± 2) days, as hybrid fertility parameters can vary with age, with a marked decline after 20 weeks (WIDMAYER *et al.* 2020). We performed regression analyses of age and fertility for each *Prdm9* allele separately and for all alleles combined and detected no apparent effect of age on fertility in the tested age ranges, with only a weak positive correlation of age on testis weights in offspring with the *mnt1^{KH}* and *msc10* alleles (**Error! Reference source not found.**).

205-236 Were there any differences in phenotyping compared to the previous study (Mukaj et al 2020)?

The phenotyping was done essentially in the same way. However, as the mice in this study were offspring of outcrossed mice whose sires were almost entirely heterozygous for *Prdm9* alleles, we used one of the epididymis for genotyping of the *Prdm9* alleles, while the other was used to count the sperm as detailed in the Material and Method section.

218-236 How many cells were scored for each male? Is this info deposited in Dryad? If so, its range must be stated here.

We had already deposited this information in the Dryad repository for each individual, but we fully agree with the Reviewer that the information should also be supplied in the main text. We have remedied this prior oversight and added a summary of the information as follows:

We evaluated 48-113 of such pachynemas for asynapsis in each individual, scoring each HORMAD2 stained element (excluding sex chromosomes) as one asynapsis event. We determined the percentage of cells with asynapsis (and collected all data in the linked Dryad Repository).

337 info redundant with 63,291,477,480,955...

We fully agree with the reviewers, and have removed some of the redundancy, while still maintaining relevant context as much as possible.

410-412 references missing, initiation unspecified
and

413-415 need to distinguish divergence in PRDM9-binding domain and PRDM9-binding sites

In response to some of the previous comments and to characterize in more detail, the differences in the divergence in the PRDM9-binding domain and the DNA motifs that the domain is predicted to bind, we have included a cartoon of the PRDM9-binding domain in

Fig3A, and several additional analyses on predicted DNA binding motifs and their genome-wide positioning.

503-504 unclear what is meant by "clustering of hybrid sterility alleles" - clustering by subspecies of ZFN alleles co-inducing hybrid sterility?

We have rephrased this paragraph, it now reads:

The phylogenetic analyses further support a scenario in which MUS and DOM have split recently and are still speciating but do not reveal any apparent clustering of alleles co-inducing hybrid sterility by subspecies. Admittedly, no evidence was found to support the idea that the hybrid sterility susceptible alleles (*msc1*, *msc1*, *msc5*, *dom2*, *dom3*, and *dom5*) belong to the evolutionary oldest ones closest to the common ancestor. On the contrary, the *msc1*, *msc2*, and *dom3* alleles are the most distal, sitting on the most distant branch of the phylogenetic tree (**Error! Reference source not found.** and **Error! Reference source not found.**).

506-507 subject missing?

Thank you for your comment, we have rephrased this paragraph and included a subject:

To investigate whether newly identified wild *Prdm9* alleles induce hybrid sterility in intersubspecific crosses, we adapted the crosses used in the laboratory models of hybrid sterility to eliminate possible variation due to the *Hstx2* modifier.

525-526 Additional explanation need, why was this outcome expected- hybrid sterility is defined as occurring only in offspring.

We apologize for lacking an explanation here, we were refereeing to expected inter-individual variation in the fertility parameters of mice, even outside of the context of hybrid sterility. We have rewritten this paragraph entirely to clarify our expectations, it now reads:

Since these fertility parameters differ with the genetic background (WIDMAYER *et al.* 2020), we first looked into the physiological variation of the outcrossed source populations of wild mice as a control, which we compared with inbred mice (**Error! Reference source not found.**).

588 "In reciprocal orientation": do you mean "To test the reciprocal orientation"?

We apologize for not being entirely clear, and have rephrased the sentence accordingly.

We also performed an analogous cross in reciprocal orientation for MUS alleles, where (PWD × wild MUS) F₁ hybrid females were crossed with B6 males.

605-607 this requires expanded explanation of the genomic modifiers, as it seems to contradict the previous sentence [603-605]

We apologize for not being entirely clear, and have rephrased the paragraph accordingly.

Siblings that inherited the *Prdm9* alleles *msc6*, *msc7*, *msc8*, or the *mmt1* of MUS or DOM all displayed fertility phenotypes within the physiologically normal range. In contrast, their brothers with the allelic combination *msc1/dom2* were sterile (**Error! Reference source not found.**), with testes' weight and sperm count comparable to (PWD × B6) F₁ males with the same allelic combination. These results suggest that mice from MUS populations with different *Prdm9*

alleles may share the same pattern of erased PRDM9^{msc1} (PWD) binding motifs, implicating uncoupling of the erased PRDM9 binding motifs and PRDM9 zinc finger arrays at the population level. Furthermore, the partially wild-derived outbred background does not appear to carry additional major genetic modifiers that would prevent sterility *per-se*.

607-608 This conclusion is only speculative. *Prdm9* *msc1* and *msc6* are not on the same background. Even taking into the account the co-segregation of the *Prdm9* alleles with fertility in [outcrossed] hybrids, variance in the *Prdm9* besides ZFN exon 12 has not been assessed.

We agree fully with the reviewer, and apologize for the confusion. We had meant to imply is that in this particular experiment full sibling littermates would show either sterility or fertility, cosegregating with the paternal *Prdm9* allele, either the *Prdm9*^{msc1} allele from the PWD grandmother, or *Prdm9*^{msc6} from the wild MUS grandfather, together with a heterogeneous recombined wildMUS/PWD genome. Our observation is that neither any variance in *Prdm9* besides ZNF exon 12, nor variation in the portion of unique genome of each individual F₁ hybrid offspring inherited from the wild mouse grandparent prevented hybrid male sterility *per se*.

We have now removed this statement and rephrased the paragraph completely.

612-615 Again speculation, as the *Cst* and *msc11* ZFN alleles have not been tested in the same system with otherwise 100% genetic background

We apologize for the confusing presentation as we had never intended to imply that there would be a 100% genetic background. We have completely removed this statement as suggested by the reviewer.

638 "wild MUS allele": would it be better "wild MUS *Prdm9* allele" ?

Thank you for this comment, we fully agree with the reviewer and have rephrased the statement accordingly.

Regardless of these X-chromosomal haplotypes, all F₁-hybrids inheriting any wild MUS *Prdm9* alleles from their mothers were fertile, whereas siblings inheriting the *msc1* alleles were sterile.

647 average sperm count must be given instead of "traces of sperm"

We fully agree with the reviewer and have now added the average sperm counts; the sentence has been rephrased accordingly.

An exception to this rule was the *Prdm9*^{msc1/dom2} male offspring, whose mothers were *t*-haplotype carriers, which produced an average of 2.7 M spermatozoa.

647-654 was the transmission ratio distortion checked for these t-haplotype mothers [separately from those having two "fertility" alleles of *Prdm9*]?

Thank you very much for this valuable comment. We had indeed also checked for transmission ratio distortion in t-haplotype mothers, which showed even transmission. We have now included the following statement:

We also performed an analogous cross in reciprocal orientation for MUS alleles, where (PWD × wild MUS) F₁ hybrid females were crossed with B6 males. In the female germline, the

mmt1 allele showed even transmission (two-tailed exact binomial probability, $P = 0.136$), confirming the male-specific *t*-haplotype transmission distortion (LYON 2003).

660-661 specify heterogeneous wild genomic

Thank you for this comment. We have eluded more strongly to the outcrossed nature of the mice we have used, also in response to server other reviewer comments. We have furthermore rephrased the sentence to clarify that we mean the wild-derived outbred genetic background.

Furthermore, the partially wild-derived outbred background does not appear to carry additional major genetic modifiers that would prevent sterility *per-se*.

664 "segregated purely" should read "co-segregated"; this experiment does not exclude other genetically linked variation

Indeed, we fully agree with the reviewer's comment, and have rephrased the entire paragraph accordingly.

If a given MUS population with an unrelated "fertility" *Prdm9* allele carries the same *msc1* binding motif as PWD mice, then the fertility of (PWD x MUS) x B6 hybrids would segregate according to *Prdm9*; *Prdm9^{msc1}* males would be sterile, and their *Prdm9^{MUS}* siblings would be fertile. However, if the given wild MUS genome does not carry any PRDM9 binding site erasure or the erased motifs match an unrelated *Prdm9* allele, then *Prdm9^{msc1}* and *Prdm9^{MUS}* male progeny should be fertile. The same assumption can be made for the outcome of PWD x (B6 x DOM) testcross to test the coupling of eroded PRDM9 binding motif with corresponding PRDM9 allelic zinc finger domains.

685 *msc1* is also present in the PWK strain

Indeed, we fully agree with the reviewer and apologize for the oversight. Due to the addition of comparative analyses of DNA binding motifs per the reviewers' suggestions, and the resulting major restructuring this part of the manuscript, the original paragraph has however since been removed.

711-712 I consider this expected rescue unlikely, because (PWD x B6)F1 hybrids with deleted dom2 allele (*Prdm9*<*msc14/tm1*>) display very low SC

This paragraph has been substantially rewritten, and the sentence in question has been removed.

719 unexplained abbreviation [LD]

We apologize for the oversight . The abbreviation (LD) is now explained as follows:

The observation that recombination maps based on linkage disequilibrium (LD) analyses revealed significant overlap of historical hotspots of the AKH population with contemporary *msc1* activated hotspots, detected using DMC1 Chromatin Immunoprecipitation and sequencing (ChIP-Seq) in the PWD strain.

727-8 "show reduced fertility": do you mean "show reduced fertility of their hybrids"?

This paragraph has been substantially rewritten, and the sentence in question has been removed.

731-34 unclear purpose of this sentence

We fully agree that this sentence had no obvious purpose and have removed it as suggested by the Reviewer

744-751 Similar statement as in 607-615: I do not see results in this paper showing that these alleles are "on the same background", only from the same population. To exclude background effect/s, resequencing of the LD region containing the ZFN domain would have to be performed in the KH population, as well as expression analysis of *Prdm9*.

We agree with the reviewer on this and several of their other points regarding the use of the term "on the same background". Throughout the manuscript, we have now rephrased what we meant.

1) the mice are indeed individuals that differ substantially in their genomic background, with individual difference occurring even between littermates that each possess a different combination of alleles from their parents, due to having genomes that have been reshuffled through meiosis, particularly pertaining to the proportion of the genome inherited from the wild mouse parent that has extensive genetic variation between homologous chromosomal haplotypes.

2) If a given MUS population with an unrelated "fertility" *Prdm9* allele carries the same *msc1* binding motif as PWD mice, then the fertility of (PWD x MUS) x B6 hybrids would segregate according to *Prdm9*; *Prdm9^{msc1}* males would be sterile, and their *Prdm9^{MUS}* siblings would be fertile. However, if the given wild MUS genome does not carry any PRDM9 binding site erasure or the erased motifs match an unrelated *Prdm9* allele, then *Prdm9^{msc1}* and *Prdm9^{MUS}* male progeny should be fertile. The same assumption can be made for the outcome of PWD x (B6 x DOM) testcross to test the coupling of eroded PRDM9 binding motif with corresponding PRDM9 allelic zinc finger domains.

751 are you saying SKE and PWD are equal on chromosome 17?

We apologize of the confuseion, as we have never meant to imply that SKE and PWD may be equal on chromosome 17, and have since substantially rewritten the paragraph, and the sentence in question has been removed.

744-751 These statements should be removed or substantiated by sequencing/expression/knock-in experiments.

These paragraphs have been substantially rewritten and we have since deleted these statements as suggested by the Reviewer

755-765 is this supported by the frequencies of *Prdm9* alleles in wild mouse populations?

Thank you for pointing out the lack of clarify in this abstract, as indeed, the ZNF-encoding exon is one of the most rapidly evolving loci in Metazoan, and there have been more than 150 alleles of *Prdm9* identified in wild mouse populations to date. We have now amended the paragraph to include this important information.

While the data on *Prdm9* polymorphism in the wild house mouse populations are accumulating, and indeed the *Prdm9* genes show remarkable natural allelic divergence, with more than 150 alleles having been found in mouse populations to date (BUARD *et al.* 2014; KONO *et al.* 2014; VARA *et al.* 2019; MUKAJ *et al.* 2020), little is known about their DNA binding motifs and their degree of erosion.

777 "We are grateful to Heike Harre for fertility phenotyping..." This should be either more specific or this person should be among authors, because a substantial part of the paper is about fertility phenotyping.

Heike Harre is a laboratory technician and animal caretaker who was of substantial help with sperm counting, and we had indeed offered co-authorship. However, Ms Harre (as well as Ms Thomsen) both graciously declined on the basis that they see their role in the technical support of the data generation, but that they were supervised (by LOH and KFNA) while doing so, and had no part in the conception, design or the analysis of their data, and have not been involved in the interpretation of the data they generated or manuscript preparation. We have now rephrased this sentence to read the following:

We are grateful to Heike Harre for help with sperm counting.

956 the color code is not explained in Fig.S2 regarding Fig.1, is it in Fig.1 also by chemical properties?

Thank you very much for this comment, we have rephrased the Figure Legend to explain better, what the color coding meant, also in response to Reviewer #2. The color code is not by chemical properties, but simply to visually distinguish different amino-acids by color, on a color-blind scale.

Please note that the original Fig1 you refer to is now Fig3, and the revised Figure Legend now reads:

Figure S2: Types of Cysteine-2-Histidine-2 (C₂H₂) ZNFs found in PRDM9 MUS and DOM in this study (MUKAJ *et al.* 2020). We highlighted the Cysteines and Histidine residues in yellow and shaded all variable amino acids in gray, distinguishing between amino acids by color. As amino acids in positions -1, 3, and 6 of the alpha-helix are those responsible for the DNA-binding specificity of a given ZNF (as show in the cartoon of **Error! Reference source not found.A**), we used them as acronyms on the right. Boxes of the same color characterize the hypervariable positions of each C₂H₂ zinc finger in **Error! Reference source not found.** and in the phylogenetic analyses in **Error! Reference source not found.**

images:

Fig.S1 MMD MMM code is used instead of DOM MUS as stated in Legend

We have now changed the code in Fig S1 from MMD MMM to DOM MUS

Fig.S5: purple color is very tough to see

Thank you very much for this comment, we have now changed the color scheme to a brighter scheme with higher visibility, also in response to the comment made by Reviewer #1.

Fig.S7-S8: probability stars missing

Thank you for this comment. Instead of adding probability stars to the Figure, we have instead added the p-values, in response to the request made by Reviewer #1.

Minor:

Thank you for the very thorough revision, and spotting these small mistakes. We appreciate that the devil is sometimes in the detail and have taken all minor suggestions into account.

42 etc. "Our data": better "Our results" [data are just numbers, one needs statistics]

Thank you for this comment, we have changed the term to now read as follows:

Our results corroborate the gradual decline of *Prdm9*-controlled hybrid sterility as the PRDM9 ZNF domain diverges.

76 "PWDxB6": PWD and B6

and

116 "(B6 X PWD)": "(B6 x PWD)" [cross sign, not a letter]

Thank you for spotting this oversight. We have now replaced the letter "x" in crosses with the special characters "x" replacing all of the previous letter x

140, 152, 156 etc. "t" in "t-haplotype/s" should be in italics

Thank you for this comment, the t in *t*-haplotypes is now italicized throughout

651 "P > 0.0001": "P < 0.0001"?

Thank you for spotting this mistake, we have corrected the symbol

322,324... why are the references italicized?

Thank you for spotting this oversight. The italicizing of references appears to have been a formatting error of the referencing software, it has now been fixed.

364 "T/tp⁴" "T" and "tp⁴" should be in italics, "p⁴" in superscript

Thank you for this comment. We have changed the strain identifier to the correct nomenclature, it now reads: *T/tp⁴*

411 etc. DOM MUS are sometimes in italics, sometimes not - please unify

We thank the reviewer for this comment, and have now unified DOM MUS throughout the document to not be in italics.

723-4 "Suppose Individual": "Suppose that individual" ?

Thank you for this comment, "suppose that individual" would have been more fitting. However due to the rewriting of the paragraph in question, this sentence has been removed.

October 2, 2023

GENETICS-2023-306479

Natural variation in the zinc-finger-encoding exon of *Prdm9* affects hybrid sterility phenotypes in mice

Dear Dr. Odenthal-Hesse:

Two experts in the field have reviewed your revised manuscript, and I have read it as well. I am pleased to inform you that, with substantial revisions to improve clarity of the narrative, your study is potentially suitable for publication in GENETICS.

However, although both Reviewers express enthusiasm for the findings presented, both Reviewer 4 and I struggled to understand several areas of the manuscript. Our comments and concerns about readability and clarity must be carefully addressed in your revision (particularly those of Reviewer 4, who has more expertise than I do in this area). You can read our reviews at the end of this email.

It is most important that you clarify the hypotheses being tested in each section of the study, and I recommend using illustrations to help the reader understand the hypotheses or models being addressed, and the approaches taken. Illustrations of the crosses performed could be extremely helpful for communicating the experimental approach and predicted potential outcomes. Illustrations and narrative improvements that clarify what potential outcomes could occur and how each outcome would be interpreted will help the reader understand the experimental question(s) being addressed and how the data speak to a particular experimental question.

Reviewer 4 also raises an important major critique that your study uses sperm count and testis weights as proxies for asymmetric *Prdm9* binding without first establishing (via sequence analyses) that these phenotypes are reliable proxies for symmetric/asymmetric *Prdm9* binding. If you agree that this is a valid concern, please directly and fully discuss this weakness of the analysis in your revised manuscript so that the reader has a full picture of the data and possible interpretations that can be drawn.

We look forward to receiving your revised manuscript. It will likely be re-reviewed by at least one expert in the field.

Please let the editorial office know approximately how long you expect to need for revisions.

Upon resubmission, please include:

1. A clean version of your manuscript;
2. A marked version of your manuscript in which you highlight significant revisions carried out in response to the major points raised by the editor/reviewers (track changes is acceptable if preferred);
3. A detailed response to the editor's/reviewers' comments and to the concerns listed above. Please reference line numbers in this response to aid the editors.

Additionally, please ensure that your resubmission is formatted for GENETICS.

<https://academic.oup.com/genetics/pages/general-instructions>

Follow this link to submit the revised manuscript: Link Not Available

Sincerely,

Amy MacQueen
Associate Editor
GENETICS

Approved by:
Jeff Sekelsky
Senior Editor
GENETICS

Reviewer #2 (Comments for the Authors (Required)):

The revised version was very attentive to the critiques. This is a well-constructed paper.

Reviewer #4 (Comments for the Authors (Required)):

Prior work has established the importance of Prdm9 in hybrid male sterility using laboratory crosses between intersubspecific inbred mouse strains. Curiously, however, wild-caught intersubspecific house mice exhibit low levels of sterility, begging the question as to whether Prdm9 contributes to hybrid sterility in the wild. To address this question, AbuAlia et al. leverage the Prdm9 diversity present in outbred populations descended from wild-caught *M. m. domesticus* and *M. m. musculus* mice to assess male reproductive parameters in hybrids with controlled Prdm9 and Hstx2 alleles on otherwise diverse genetic backgrounds. Through a clever series of crosses, they show that the dom2 Prdm9 allele is associated with a reduction in hybrid fitness, irrespective of genetic background.

Major:

Overall, this paper summarizes an enormous body of work and adds intriguing observations to the puzzle of Prdm9 and its role in hybrid male sterility and genome evolution. Enthusiasm for this work is, however, diminished by what I judge to be poor composition. The text is redundant in many places, content is presented in a disorganized fashion, and hypotheses are not always clearly articulated. I also struggled to understand the crossing schema employed from reading the text alone and believe a conceptual figure illustrating the various crosses and the associated Prdm9/Hstx2 genotypes would be a very valuable addition to aid causal readers of this article (especially those who are not steeped in the Prdm9 literature!).

I also struggled to appreciate the interpretation of findings in the section entitled "The relationship between Prdm9 genotype and wild-derived outbred genetic background". Again, a schematic may help the reader parse the various hypotheses being tested here. The experimental test presented in this section relies on sperm count and testis weights as read outs for asymmetric Prdm9 binding without establishing that these phenotypes are reliable proxies for binding asymmetry in the first place. To me, a demonstration of hotspot erosion would seem to require sequence-based analyses.

Minor:

There are a large number of typos throughout the manuscript -- too many to enumerate individually. Some careful editing is needed to clean up the text.

Line 199: unclear what is meant by this clause. How can a single allele be the "most diverse"? Do you mean "divergent", and if so, divergent from what?

Line 255: body weight is not a fertility parameter.

Line 325-326: How many Prdm9 alleles had less than 12 ZNFs?

Lines 419-423: this duplicates information presented in the Methods.

Line 479: C57Bl6/J C57BL/6J

Line 484-486: The content communicated in this sentence has already been presented. This sentence could be deleted to streamline the text.

Lines 499-501: The testis weight data (bottom left panel) appear to be bimodal, with two clusters corresponding to high and low testis weight groupings. Thus, it is not appropriate to fit a single regression to the data. The suggestion presented by reviewer 1 that testis weights should be normalized to body weight remains valid, in my opinion.

Lines 532-544: much of this content echoes what is in the Methods

Line 603-604: While I agree that the discovery of multiple haplotypes in this region is consistent with possible historical recombination, it is also potentially the case that high rates of genome instability at microsatellite markers has led to recurrent mutations that masquerade as recombination. This alternative should be acknowledged.

Associate Editor Comments:

Line 66. and others: Sometimes "Helix" is capitalized, and sometimes not. I do not think it should be capitalized.

Line 124. Please clarify to the reader (perhaps with an illustration if necessary) why asymmetrical hotspots pose a fertility problem.

Line 126. Please clarify here and later in the manuscript why "oligogenic" control is concluded when modifying a single gene can restore fertility. Why not conclude that hybrid sterility is under the control of a single gene?

Line 146. Double check: is "synapsis" meant to be "asynapsis"?

Line 149. Please clarify the information conveyed in this sentence. What is meant by hotspot erosion being exacerbated, and what is meant by "strain-specific allele" in a hybrid context - is it referring to two parental alleles? And what is meant by "activates all hotspots" (particularly if hotspots are eroded)?

Line 199. "diverse" should be replaced by "divergent" (and see Reviewer comment regarding clarification of what the comparison group is when you say "most divergent")

Line 203. Typo - insert "the" between in and DOM/MUS.

Lines 395-399. Something is wrong with the structure of these sentences. Check and fix.

Line 449. Remove "Even though" from the sentence.

Line 462 and throughout the rest of the section: amino acid residues should not be capitalized.

Line 517. Define "TW" and "SC" (earlier when they are spelled out)

Line 533. SYCP3 is a protein that decorates the meiotic chromosome axis and is not a reliable readout for SC assembly (in mutants missing the SYCP1 or TEX12 SC central region structural components, SYCP3 staining is intact). The reason I believe you immunostained for SYCP3 is to identify pachytene stage nuclei, which show continuous SYCP3 staining on chromosome axes. Thus, you should remove "when the synaptonemal complex is fully formed" from line 532 (because SYCP3 staining does not indicate this unless you use it to look closely at whether the axes are intimately aligned along their entire lengths). Remove "(when the synaptonemal complex is still forming)" and "when the synaptonemal complex begins to disassemble.....)" from the next sentence, because these lines are misleading and also unnecessary for the purpose of your experiment.

Line 542. Unlike the mention of synaptonemal complex assembly to stage nuclei above, it is critical to explain why HORMAD2 immunolocalization is used here.

Line 611 and throughout text: It was my belief that genetic loci/alleles (unlike protein) are italicized. When referring to "Prdm9 alleles" or "msc1" should Prdm9 and msc1 be italicized?

Lines 607 plus/minus. As mentioned by a reviewer, I found myself wanting to see an illustration of the crosses and phenotypic outcomes, in order to help clarify the experiments and the interpretation.

Lines 647-654. This section is completely unclear; please rewrite in a manner that ensures readers who are not specialists can grasp what the issues are and how the approaches attempt to solve the problems. Define what Hamming distance is in a way that is understandable, and what the HMMER bit score means.

Line 725 and other: "enquire" should be "inquire".

Response to Reviewers

Abualia *et al.* Natural variation in the zinc-finger-encoding exon of *Prdm9* affects hybrid sterility phenotypes in mice

GENETICS-2023-306479

We would like to thank the anonymous peer-reviewers for their detailed comments and thorough review to improve our manuscript. We are excited to read your overall enthusiasm and recognition of the improvement in clarity and, of course, the general recommendation that our manuscript is suitable for publication once the minor comments are addressed.

We have carefully incorporated all of the referees' minimal suggestions for finalizing our manuscript to the best of our knowledge and hope that our manuscript is now ready for publication. As before, we have highlighted all amendments using tracked changes,

Below, we provide a detailed, point-by-point response to the specific comments/questions/issues raised in the reviews. The original Reviewer's comments are in *black text*, our responses are in blue.

Reviewer #2 (Comments for the Authors (Required)):

The revised version was very attentive to the critiques. This is a well-constructed paper.

We are grateful for the continued enthusiasm for this manuscript, and the recognition of the amount of work that we have put into improving the clarity of the manuscript.

Reviewer #4 (Comments for the Authors (Required)):

Prior work has established the importance of *Prdm9* in hybrid male sterility using laboratory crosses between intersubspecific inbred mouse strains. Curiously, however, wild-caught intersubspecific house mice exhibit low levels of sterility, begging the question as to whether *Prdm9* contributes to hybrid sterility in the wild. To address this question, AbuAlia *et al.* leverage the *Prdm9* diversity present in outbred populations descended from wild-caught *M. m. domesticus* and *M. m. musculus* mice to assess male reproductive parameters in hybrids with controlled *Prdm9* and *Hstx2* alleles on otherwise diverse genetic backgrounds. Through a clever series of crosses, they show that the *dom2 Prdm9* allele is associated with a reduction in hybrid fitness, irrespective of genetic background.

Major:

Overall, this paper summarizes an enormous body of work and adds intriguing observations to the puzzle of *Prdm9* and its role in hybrid male sterility and genome evolution. Enthusiasm for this work is, however, diminished by what I judge to be poor composition. The text is redundant in many places, content is presented in a disorganized fashion, and hypotheses are not always clearly articulated. I also struggled to understand the crossing schema employed from reading the text alone and believe a conceptual figure illustrating the various crosses and the associated *Prdm9/Hstx2* genotypes would be a very valuable addition to aid causal readers of this article (especially those who are not steeped in the *Prdm9* literature!).

Thank you for the critical comments. We tried our best to make the text more comprehensive and less confusing and deleted the possible redundancies. Two conceptual figure with the crossing schemes were added.

I also struggled to appreciate the interpretation of findings in the section entitled "The relationship between *Prdm9* genotype and wild-derived outbred genetic background". Again, a schematic may help the reader parse the various hypotheses being tested here. The experimental test presented in this section relies on sperm count and testis weights as read outs for asymmetric *Prdm9* binding without establishing that these phenotypes are reliable proxies for binding asymmetry in the first place. To me, a demonstration of hotspot erosion would seem to require sequence-based analyses.

Hotspot erosion in our laboratory model of hybrid sterility (PWD x B6)F1 was described by Davies *et al* 2016 and Smagulova *et al.* 2016. Ideally, DMC1 ChIP-seq should be done in our current crosses with wild mice, but it is technically extremely difficult and with outcrossed wild mice almost impossible.

We have fundamentally rewritten this part, including the title. The introductory section explains the logic of the experiment as follows:

Role of the wild-derived outbred genetic background in *Prdm9*-driven meiotic arrest

The hypothesis that *Prdm9*-driven hybrid sterility can be explained through the asymmetric erosion of PRDM9 binding motifs was based on the documented erosion of MUS PRDM9^{msc1} binding sites in the PWD genome and DOM PRDM9^{dom2} sites in the B6 genome (Davis et al. 2016). This hypothesis explains the F₁ hybrid sterility resulting from heterozygosity for functional PRDM9 binding sites. However, while *Prdm9* can be considered one of the most polymorphic mammalian genes (Oliver et al. 2009; Grey et al. 2018), almost nothing is known about the occurrence of PRDM9 binding site erosion in mouse populations or how many alleles have generated significant erosion of their binding sites and whether these eroded sites are stable in wild mouse populations. Following the asymmetry hypothesis, a 'fertility' *Prdm9* allele from a male with no significant erasure of PRDM9 binding sites in the MUS population should produce fertile progeny in (PWD × MUS) × B6 hybrids because erasure of PRDM9^{msc1} sites in the PWD part of the genome alone is not sufficient to disrupt meiosis in hybrids. However, if the MUS genome with the 'fertility' *Prdm9* allele carried eroded *msc1* binding sites, then the fertility of (PWD × MUS) × B6 hybrids would segregate according to *Prdm9*; the *Prdm9*^{msc1} carrying males would be sterile, and their *Prdm9*^{MUS} siblings would be fertile. The same assumption can be made for the outcome of PWD × (B6 × DOM) testcross to test the coupling of eroded PRDM9 binding motif with corresponding PRDM9 allelic zinc finger domains in DOM populations. To test these alternatives, (...)

Minor:

There are a large number of typos throughout the manuscript -- too many to enumerate individually. Some careful editing is needed to clean up the text.

We thank the reviewer for this comment. We have now carefully proofread the submission, also applying the Proofreading software "Grammarly" to prevent typos.

Line 199: unclear what is meant by this clause. How can a single allele be the "most diverse"? Do you mean "divergent", and if so, divergent from what?

The sentence has been corrected to read:

Naturally occurring chromosome 17 haplotypes, the *t*-haplotypes (SILVER 1985), also strongly influence male fertility in wild mice. They consist of 30 Mb of introgressed sequence transferred from an unidentified *Mus* ancestor in the *Mus musculus* subspecies over one million years ago (HAMMER AND SILVER 1993) and encompass the *Prdm9* gene (TRACHTULEC et al. 2008), with the allele most divergent from all other *Prdm9* alleles identified to date (Kono et al. 2014).

Line 255: body weight is not a fertility parameter.

You are absolutely correct; we have rephrased the sentence to now read the following:

We collected body weight (BW) and two fertility parameters, paired testes weight (TW) and spermatozoa released from epididymal tissues, counted in Million/ml (SC).

Line 325-326: How many *Prdm9* alleles had less than 12 ZNFs?

Three different *Prdm9* alleles had less than 12 ZNF, *msc11* from Kazakhstan (this study), *dom5* from Mukaj et al and the allele associated with *t*-haplotype carriers in all populations.

Lines 419-423: this duplicates information presented in the Methods.

Thank you for this comment, we amended the Methods as well as the Information in the Results and Discussion to avoid redundancy. The relevant part now reads:

We screened all populations for individual *Prdm9* alleles by sequencing exon12 of *Prdm9* containing the minisatellite coding for the C₂H₂ zinc-finger domain of PRDM9 as described in (Buard et al. 2014; Kono et al. 2014). The amino-

acid variation between individual zinc fingers is shown in (**Error! Reference source not found.** and **Error! Reference source not found.**).

Line 479: C57Bl6/J \diamond C57BL/6J

We have now corrected the strain name to always read C57BL/6J

Line 484-486: The content communicated in this sentence has already been presented. This sentence could be deleted to streamline the text.

Thank you for this comment, we have deleted this sentence, and also rephased the next sentence to streamline the text.

Lines 499-501: The testis weight data (bottom left panel) appear to be bimodal, with two clusters corresponding to high and low testis weight groupings. Thus, it is not appropriate to fit a single regression to the data. The suggestion presented by reviewer 1 that testis weights should be normalized to body weight remains valid, in my opinion.

Thank you for your valuable comment. We totally agree, as the differences in testis weights are driven by *Prdm9* in the example you give, the distribution appears bimodal (sterile and fertile), and as the same is true whether or not testis weight or testis weight normalized to body weight was used, we have now exchange testis weight to testes weight normalized to body weight in all figures of the manuscript. As now the comparative prior Supplementary Figure S8) is redundant, we have removed it from the Supplementary Data.

Lines 532-544: much of this content echoes what is in the Methods

Most of this text was deleted.

Line 603-604: While I agree that the discovery of multiple haplotypes in this region is consistent with possible historical recombination, it is also potentially the case that high rates of genome instability at microsatellite markers has led to recurrent mutations that masquerade as recombination. This alternative should be acknowledged.

You are entirely correct, that microsatellites show high rates of genome instability as they evolve by replication slippage. To address your valid point that recurrent mutations might masquerade as recombination, we have now added the following sentence "However, it cannot be excluded that high rates of genome instability at minidatellites and microsatellite markers have led to recurrent mutations masquerading as recombination."

Associate Editor Comments:

Line 66. and others: Sometimes "Helix" is capitalized, and sometimes not. I do not think it should be capitalized.

Thank you for this comment, we have now unified "helix" to not be capitalized throughout.

Line 124. Please clarify to the reader (perhaps with an illustration if necessary) why asymmetrical hotspots pose a fertility problem.

The following sentence was added: "As a result, DNA DSBs initiated at asymmetric sites are difficult or impossible to repair, activating the DNA repair checkpoint and/or disrupting homolog pairing and synapsis, both of which lead to meiosis arrest."

We hope that this explanation can help the reader to understand the proposed mechanism of sterility.

Line 126. Please clarify here and later in the manuscript why "oligogenic" control is concluded when modifying a single gene can restore fertility. Why not conclude that hybrid sterility is under the control of a single gene?

Line 146. Double check: is "synapsis" meant to be "asynapsis"?

We indeed were referring to the restoration of synapsis, but agree that this sentence was confusing in the context of asynapsis. We have therefore rephrased the sentence to clarify what we mean:

Asynapsis was shown to operate *in-cis*, depending on the increased heterozygosity of homologs from evolutionarily divergent subspecies. Introducing at least 27 Mbs of sequence homology belonging to the same subspecies (con-specific homology) rescued the asynapsis of a given autosomal pair (GREGOROVA *et al.* 2018).

Line 149. Please clarify the information conveyed in this sentence. What is meant by hotspot erosion being exacerbated, and what is meant by "strain-specific allele" in a hybrid context - is it referring to two parental alleles? And what is meant by "activates all hotspots" (particularly if hotspots are eroded)?

We have rephrased the entire paragraph to clarify what we mean, and also included a figure to explain our hypothesis, specifically, the sentence was changed as follows: "Complete sterility has been observed only in F₁ hybrids heterozygous for *Prdm9* alleles *dom2* and *msc1* (DAVIES *et al.* 2016; SMAGULOVA *et al.* 2016; MUKAJ *et al.* 2020)."

Line 199. "diverse" should be replaced by "divergent" (and see Reviewer comment regarding clarification of what the comparison group is when you say "most divergent")

We fully agree and have rephrased the sentence as follows: "Naturally occurring chromosome 17 haplotypes, the *t* - haplotypes (SILVER 1985), also strongly influence male fertility in wild mice. They consist of 30 Mb of introgressed sequence transferred from an unidentified *Mus* ancestor in the *Mus musculus* subspecies over one million years ago (HAMMER AND SILVER 1993) and encompass the *Prdm9* gene (TRACHTULEC *et al.* 2008), with the allele most divergent from all other *Prdm9* alleles identified in *Mus musculus* to date (Kono *et al.* 2014)."

Line 203. Typo - insert "the" between in and DOM/MUS.

We have inserted "the"

In summary, the low incidence of sterile wild-mouse hybrids in the DOM/MUS natural hybrid zone (TURNER *et al.* 2012) together with the reported large number of hybrid sterility loci in intersubspecific backcrosses and intercrosses contrasts with the F₁ hybrid sterility model based on the *Prdm9* allelic incompatibility, *Hstx2*, and background heterozygosity in PRDM9 binding sites.

Lines 395-399. Something is wrong with the structure of these sentences. Check and fix.

We have rephrased this sentence and split it into several sentences. The section now reads as follows: Mice were initially caught by (Harr *et al.* 2016), founders of the MUS population AKH were initially trapped in Almaty, Kazakhstan (43°16'N, 76°53'E), and founders of DOM populations came from three different locations, for the AHI population AHI from the city of Ahvaz, Iran (31°19' N, 48°42' E), for population MCF in from the Massif-Central region in France (45°32'N, 2°49'E), and for the CBG population founders were trapped in the Cologne-Bonn region in Germany (50°52'N, 7°8'E). All populations have been housed and maintained as outcrosses for many generations before this study (see Materials and Methods), and these populations have maintained a much larger genomic diversity than inbred strains despite high degrees of relatedness (Lawal *et al.* 2021).

Line 449. Remove "Even though" from the sentence.

We have removed "even though"

Line 462 and throughout the rest of the section: amino acid residues should not be capitalized.

Thank you for this valuable comment, we have now corrected the amino acid capitalization, the section now reads as follows:

For example, PRDM9^{msc11} closely resembled two variants from the CAS subspecies, the classical PRDM9^{cs11} variant, which possesses serine at position -1 of the alpha-helix of the 8th ZNF (Parvanov *et al.* 2010), instead of the asparagine seen in PRDM9^{msc11}. A CAS trapped in Nowshahr, Iran, Ca1 (Kono *et al.* 2014), also differs only by a single amino-acid substitution in position 6 of the alpha-helix of the 6th ZNF, where Ca1 possesses glutamine instead of the lysine seen in PRDM9^{msc11}.

Line 517. Define "TW" and "SC" (earlier when they are spelled out)

Mice with *Prdm9* alleles that were closely related to previously identified hybrid sterility alleles showed reduced sperm counts (SC) and low paired testes weights, normalized to body weight (TW/BW) that were associated with high asynapsis rates of homologous chromosomes in meiosis I and early meiotic arrest (Mukaj *et al.* 2020).

Line 533. SYCP3 is a protein that decorates the meiotic chromosome axis and is not a reliable readout for SC assembly (in mutants missing the SYCP1 or TEX12 SC central region structural components, SYCP3 staining is intact). The reason I believe you immunostained for SYCP3 is to identify pachytene stage nuclei, which show continuous SYCP3 staining on chromosome axes. Thus, you should remove "when the synaptonemal complex is fully formed" from line 532 (because SYCP3 staining does not indicate this unless you use it to look closely at whether the axes are intimately aligned along their entire lengths). Remove "(when the synaptonemal complex is still forming)" and "when the synaptonemal complex begins to disassemble.....)" from the next sentence, because these lines are misleading and also unnecessary for the purpose of your experiment.

We fully agree with the Editors comment, and have amended the section as follows:

To determine the frequency of pachytene spermatocytes with one or more asynapsed bivalents, we immunostained synaptonemal complexes using synaptonemal-complex protein 3 (SYCP3) antibody. Additionally, we assessed phosphorylated histone γ H2AX, and HORMAD2 proteins, which localize to asynapsed autosomes and to the X and Y chromosomes at mid-to-late pachytene (Fernandez-Capetillo *et al.* 2003; Turner *et al.* 2005) (**Error! Reference source not found.C**).

Line 542. Unlike the mention of synaptonemal complex assembly to stage nuclei above, it is critical to explain why HORMAD2 immunolocalization is used here.

Done. See the above response.

Line 611 and throughout text: It was my belief that genetic loci/alleles (unlike protein) are italicized. When referring to "Prdm9 alleles" or "msc1" should *Prdm9* and *msc1* be italicized?

You are entirely correct. According to the MGI nomenclature for mice, both the gene name "*Prdm9*" and the allele designations "*msc1*" should be italicized. To account for this, we also italicized the composite term, f.e. „*Prdm9^{msc1}*“ when referring to a specific allele of the gene. However, when referring to the protein, we capitalized the protein name (PRDM9). In cases where we refer to a specific encoded protein variant we did not italicize the variant designation: for example referring to the variant encoded by the *Prdm9^{msc1}* allele as the PRDM9^{msc1} variant.

Lines 607 plus/minus. As mentioned by a reviewer, I found myself wanting to see an illustration of the crosses and phenotypic outcomes, in order to help clarify the experiments and the interpretation.

Thank you very much for this comment. We have now included cartoons illustrating the different types of crosses that were performed for this study.

Lines 647-654. This section is completely unclear; please rewrite in a manner that ensures readers who are not specialists can grasp what the issues are and how the approaches attempt to solve the problems. Define what Hamming distance is in a way that is understandable, and what the HMMER bit score means.

Therefore, to reflect *Prdm9* evolution more accurately, we applied an algorithm that computes Hamming Distances between minisatellite repeats. A Hamming distance is a string metric of the number of substitutions or errors needed to change one sequence into another, where all sequences of equal length are vectorized over a finite field. Here we apply it to compare *Prdm9* minisatellite 84 bp-repeat units against each other, such that not only point mutations and small indels but also within-repeat-unit processes ($w_{mut}=1$), as well as repeat-unit insertions and deletions ($w_{indel}=3.5$) and even repeat-unit duplications and slippage ($w_{slippage}=1.75$) are taken into account (VARA *et al.* 2019; DAMM *et al.* 2022).

We restricted the Hamming metric to high-confidence nucleotide repeats for a more conservative phylogenetic analysis. For each translated amino acid sequence, we first determined the bit-score, a measure of confidence in the homology of a given ZNF to the prediction model, the so-called default Hidden Markov Model (HMMER) gathering threshold. Confidence in a given ZNF is achieved with a bit score above 17.7 for the model used (PERSIKOV AND SINGH 2014), which was seen for all translated 84-bp repeats, except those coding for the first zinc fingers in the ZNF array, which we found to be conserved in all *Mus musculus Prdm9* alleles, except *mmt1*.

Line 725 and other: "enquire" should be "inquire".

Thank you for this comment, we have changed "enquire" to "inquire" throughout the manuscript.

December 8, 2023

RE: GENETICS-2023-306660

Dear Dr. Odenthal-Hesse:

I am pleased to accept your manuscript entitled "**Natural variation in the zinc-finger-encoding exon of *Prdm9* affects hybrid sterility phenotypes in mice**" for publication in GENETICS, pending minor revision.

Please submit your revision along with a brief description of how you modified the manuscript in response to the reviewer's concern (which can be viewed at the bottom of this email) that the main text should explicitly raise the caveat of not having directly assessed *Prdm9* binding. This clarification is important for rigor, and can be addressed first I believe in line 574. For example, the narrative could read "To conclude, while asymmetric *Prdm9* binding has not been directly demonstrated in our study, these results suggest..."). The caveat should again be explicitly raised in the Conclusion section of the main text, within the region of lines 754-759. I also believe that too much of the data presented in this manuscript is in supplemental files. Please move Supplemental files S4, S5, S6, S7 and S8 to the main figures (S4 would be Figure 1), as these data are core to the key experiments performed and results presented in the narrative. I welcome the authors to make additional translocations of supplemental tables and/or files to main figures if they feel it is warranted for clarity.

I expect you should be able to submit a revised manuscript within 30 days. A suitably revised manuscript will be acceptable for publication; I don't expect to send it out for review.

When revising the ms., please make an effort to shorten it, because that almost always improves a manuscript. We urge authors to heed the advice of Strunk and White: "omit needless words"¹. Follow this link to submit the revised manuscript: Link Not Available

Thank you for submitting this story to Genetics.

Sincerely,

Amy MacQueen
Associate Editor
GENETICS

Approved by:
Jeff Sekelsky
Senior Editor
GENETICS

Reviewer comments:

Reviewer #4 (Comments for the Authors (Required)):

I thank the authors for their careful attention to my prior comments. I do agree that many parts of the paper are now clarified. However, my earlier concern about the use of sperm counts and testis weights as read-outs for asymmetric *Prdm9* binding is not fully addressed in the revised manuscript. I appreciate the infeasibility of performing DMC1 ChIP-seq experiments on mice from their crosses but do think a brief mention of this caveat is deserving in the main text.

Response to Reviewers

Abualia *et al.* **Natural variation in the zinc-finger-encoding exon of *Prdm9* affects hybrid sterility phenotypes in mice**

GENETICS-2023-306660

We are pleased that our manuscript has been accepted for publication. We have carefully incorporated the last of the Editors' and referees' minimal suggestions and finalized our manuscript. As before, we have highlighted all minor changes as tracked changes.

Reviewer #4 (Comments for the Authors (Required)):

I thank the authors for their careful attention to my prior comments. I do agree that many parts of the paper are now clarified. However, my earlier concern about the use of sperm counts and testis weights as read-outs for asymmetric *Prdm9* binding is not fully addressed in the revised manuscript. I appreciate the infeasibility of performing DMC1 ChIP-seq experiments on mice from their crosses but do think a brief mention of this caveat is deserving in the main text.

We have carefully incorporated this suggestion based on the associate editor's recommendations.

Associate Editor Comments:

Please submit your revision along with a brief description of how you modified the manuscript in response to the reviewer's concern (which can be viewed at the bottom of this email) that the main text should explicitly raise the caveat of not having directly assessed *Prdm9* binding.

This clarification is important for rigor, and can be addressed first I believe in line 574. For example, the narrative could read "To conclude, while asymmetric *Prdm9* binding has not been directly demonstrated in our study, these results suggest...").

Thank you for this comment. We have changed the sentence accordingly and it now reads: To conclude, while asymmetric *Prdm9* binding has not been directly demonstrated in our study, due to the infeasibility of performing DMC1 ChIP-seq experiments on wild mouse hybrids, these results nevertheless suggest that mice from MUS and DOM populations with different *Prdm9* alleles may share the same pattern of erased PRDM9^{msc1} (PWD) and PRDM9^{dom2} binding motifs, respectively, suggesting that the erased PRDM9 binding motifs and *Prdm9* alleles are uncoupled at the population level.

The caveat should again be explicitly raised in the Conclusion section of the main text, within the region of lines 754-759.

Although our study did not directly demonstrate asymmetric *Prdm9* binding, because DMC1 ChIP-seq experiments on wild mouse hybrids are not feasible, these results nonetheless provide evidence of eroded *msc1* hotspots in five populations with fertile *Prdm9* alleles.
This was surprising, as it suggests a decoupling of the evolutionary dynamics of the PRDM9 zinc-finger domains and their binding sites.

I also believe that too much of the data presented in this manuscript is in supplemental files. Please move Supplemental files S4, S5, S6, S7 and S8 to the main figures (S4 would be Figure 1), as these data are core to the key experiments performed and results presented in the narrative. I welcome the authors to make additional translocations of supplemental tables and/or files to main figures if they feel it is warranted for clarity.

Thank you for this comment. We have moved these Supplementary figures to the main body of the text and changed the cross-referencing throughout. This also warranted the moving down of a single paragraph concerning the age correlation of hybrids.

January 5, 2024

RE: GENETICS-2023-306660R1

Dr. Linda Odenthal-Hesse
Max-Planck-Institut für Evolutionsbiologie
Evolutionary Genetics
2 August-Thienemann Str
Plön 24306
Germany

Dear Dr. Odenthal-Hesse:

Congratulations! We are delighted to inform you that your manuscript entitled "**Natural variation in the zinc-finger-encoding exon of *Prdm9* affects hybrid sterility phenotypes in mice**" is acceptable for publication in GENETICS. Many thanks for submitting your research to the journal.

I have read over your manuscript and have only one sentence re-phrasing suggestion for clarification, my suggestion is appended to the end of this message.

To Proceed to Production:

1. Format your article according to GENETICS style, as discussed at <https://academic.oup.com/genetics/pages/general-instructions>, and upload your final files at <https://genetics.msubmit.net>.
2. Your manuscript will be published as-is (unedited-as submitted, reviewed, and accepted) at the GENETICS website as an Advanced Access article and deposited into PubMed shortly after receipt of source files and the completed license to publish. Please notify sourcefiles@thegsajournals.org if you do not wish to publish your article via Advanced Access.
3. We invite you to submit an original color figure related to your paper for consideration as cover art. Please email your submission to the editorial office or upload it with your final files. You can submit a small-sized image for evaluation, and if selected, the final image must be a TIFF file 2513px wide by 3263px high (8.375 by 10.875 inches; resolution of 600ppi). Please avoid graphs and small type.

If you have any questions or encounter any problems while uploading your accepted manuscript files, please email the editorial office at sourcefiles@thegsajournals.org.

Sincerely,

Amy MacQueen
Associate Editor
GENETICS

Approved by:
Jeff Sekelsky
Senior Editor
GENETICS

note: Please add jnls.author.support@oup.com and genetics.oup@kwglobal.com (or the domains @oup.com and @kwglobal.com) to your email program's "safe senders" list. You will be contacted by both at various points during the production process.

****One final suggestion from the Associate Editor:**

I encourage the authors to modify line 519 in order to fully resolve a still lingering concern I have that the narrative implies SYCP3 is a read-out for synaptonemal complex (SYCP3 is an axis protein that remains on chromosome axes even if SC is absent; the authors can even see evidence for this in the images they present in Figure 6). I suggest line 519 be re-phrased to read: "To determine the frequency of pachytene spermatocytes with one or more asynapsed bivalents, we immunostained meiotic nuclei with antibodies against phosphorylated histone gammaH2AX and HORMAD2 proteins, which localize to asynapsed autosomes and to the X and Y chromosomes at mid to late pachytene stage (REFERENCES); we also stained these

nuclei with antibodies against the SYCP3 protein, which decorates meiotic chromosome axes."